# Could a Lower Toll-like Receptor (TLR) and NF-κB Activation Due to a Changed Charge Distribution in the Spike Protein Be the Reason for the Lower Pathogenicity of Omicron?

**DOI:** 10.3390/ijms23115966

**Published:** 2022-05-25

**Authors:** Ralf Kircheis, Oliver Planz

**Affiliations:** 1Syntacoll GmbH, 93342 Saal an der Donau, Germany; 2Interfaculty Institute for Cell Biology, Department of Immunology, Eberhard Karls University Tuebingen, 72076 Tübingen, Germany; oliver.planz@uni-tuebingen.de

**Keywords:** Omicron, spike protein, SARS-CoV-2, COVID-19, cytokine storm, NF-kappaB, Toll-like receptor (TLR)

## Abstract

The novel SARS-CoV-2 Omicron variant B.1.1.529, which emerged in late 2021, is currently active worldwide, replacing other variants, including the Delta variant, due to an enormously increased infectivity. Multiple substitutions and deletions in the N-terminal domain (NTD) and the receptor binding domain (RBD) in the spike protein collaborate with the observed increased infectivity and evasion from therapeutic monoclonal antibodies and vaccine-induced neutralizing antibodies after primary/secondary immunization. In contrast, although three mutations near the S1/S2 furin cleavage site were predicted to favor cleavage, observed cleavage efficacy is substantially lower than in the Delta variant and also lower compared to the wild-type virus correlating with significantly lower TMPRSS2-dependent replication in the lungs, and lower cellular syncytium formation. In contrast, the Omicron variant shows high TMPRSS2-independent replication in the upper airway organs, but lower pathogenicity in animal studies and clinics. Based on recent data, we present here a hypothesis proposing that the changed charge distribution in the Omicron’s spike protein could lead to lower activation of Toll-like receptors (TLRs) in innate immune cells, resulting in lower NF-κB activation, furin expression, and viral replication in the lungs, and lower immune hyper-activation.

## 1. Introduction

The novel SARS-CoV-2 Omicron variant of concern (VoC) B.1.1.529 was first detected in the context of exceptionally high infection numbers in South Africa and Botswana in November 2021, spread worldwide within a few weeks, and is now replacing all other variants worldwide, including the previously dominant Delta variant [1]. The emergence of Omicron has been associated with a dramatic increase in infection case numbers, with doubling times of few days, currently counting for far more than 90% of the cases in the USA and in most European countries, with the fastest growth reported for UK, Denmark, and France, reaching up to more than 400,000 infections per day.

In addition to the dramatically increased infectivity, Omicron has been found to largely evade therapeutic monoclonal antibodies and vaccine-induced polyclonal neutralizing antibodies after primary/secondary immunization. Mutations in the N-terminal domain (NTD) and receptor binding domain (RBD) domains of the spike protein targeted by therapeutic monoclonal antibodies correlate with significantly decreased binding and neutralization in experimental studies and in serum after primary and secondary vaccination with all tested vaccine types, including mRNA-based vaccines, BNT162b2 (BioNTech/Pfizer) and mRNA-1273 (Moderna), and even a more dramatic decrease in neutralizing activity in serums following vector-based SARS-CoV-2 vaccines, such as ChAdOx1 nCoV-19 (AZD1222, AstraZeneca), Ad26.COV2. S (Johnson & Johnson/Janssen), Gam-COVID-19-Vac (“Sputnik V”, Gamaleya National Centre of Epidemiology and Microbiology, Moscow, Russia), or the inactivated vaccine Coronavac (Sinovac, Sinovac Biotech, Beijing, China) [2,3,4]. This had been expected very soon after the sequence of the Omicron VoC became available because of the highly unusual genetic profile of Omicron with 50 genetic changes, including, exclusively, spike protein 30 amino acid substitutions, one insertion of three amino acids, and several small deletions when compared to the original Wuhan strain. These changes comprise an unprecedented sampling of mutations from earlier VoC, i.e., Alpha, Beta, Gamma, and Delta, together with other substitutions not found in any of the previous VoCs [3,5,6].

The spike protein is responsible for both the adherence of the virus to the host cells and the invasion of the virus into the host cell making the spike protein most critical for viral transmission. All vaccines available today target the spike protein and are based on the original strain first detected in Wuhan, China, at the end of 2019. Changes in the amino acid sequence in the spike protein are expected to affect both the transmissibility of the virus and the ability of the virus to evade neutralizing antibodies. Many of the substitutions and deletions in the NTD and in the RBD can meanwhile be correlated with the observed increased infectivity and transmissibility, and with the high evasion potential from therapeutic monoclonal antibodies and vaccine-induced polyclonal neutralizing antibodies. More enigmatic are the three mutations near the S1/S2 furin cleavage site which were expected to favor cleavage. Contrary to what had been expected, the observed cleavage efficacy has been found to be substantially lower in the Omicron variant compared to the Delta variant and the Wuhan wild-type virus, correlating with significantly lower TMPRSS2-dependent replication in the lungs [3,7,8,9].

A lower virus replication in lungs, together with a faster replication in the upper respiratory system, such as nasopharyngeal and bronchi, can largely explain Omicron’s greater ability for transmission between people while apparently causing less frequently acute respiratory distress syndrome (ARDS) of the lungs and systemic symptoms of COVID-19. However, the molecular mechanisms responsible for these reciprocal changes in cellular tropism of the Omicron variant regarding the upper and lower respiratory system are not sufficiently defined at this moment. Based on the data available so far, we present here a new hypothesis proposing that the changed distribution of charged amino acids in the spike protein of the Omicron variant compared to all other VoCs may disturb the recognition by innate Pattern-recognition receptors (PRRs), in particular of certain Toll-like receptors (TLR), resulting in lower activation of the NF-κB pathway and related signaling pathways, and also resulting in lower furin expression, lower viral replication in the lungs, and lower systemic immune hyper-activation.

## 2. SARS-CoV-2 and COVID-19—The Virus and the Disease

SARS-CoV-2 belongs to enveloped positive-sense, single-stranded RNA viruses, similar to the two other highly pathogenic coronaviruses, SARS-CoV and Middle East respiratory syndrome (MERS-CoV) [10]. SARS-CoV-2 binds primarily to the angiotensin-converting enzyme-related carboxypeptidase-2 (ACE2) receptor on target cells by its spike (S) protein. In addition to ACE2, additional cellular co-receptors have been identified as potential binding targets for SARS-CoV-2, including integrins, CD147, heparane sulfate, sialic acid, and neutropilin-1 [11,12]. There are two principal cellular entry routes for the virus: the highly efficient plasma membrane route and the cathepsin L-dependent endosomal entry route, dependent on whether TMPRSS2 is co-expressed with ACE2 on the host cell or not [13,14,15,16].

The S protein of SARS-CoV-2 is a class I viral membrane fusion protein that exists as a trimer, covered with 22 predicted N-glycosylation sites [17,18]. The spike glycoprotein consists of a large ectodomain, a single-pass transmembrane anchor, and a short C-terminal intracellular tail. The ectodomain contains the receptor binding S1 subunit and the S2 subunit responsible for membrane fusion. The RBD of S1 binds to ACE2 on the target cells with high affinity, whereas S2 mediates fusion between the viral and host cell membranes. The S2 subunit includes two heptad repeat (HR) regions (HR1 and HR2), the proteolytic site (S2) for the TMPRSS2 serine protease, a hydrophobic fusion peptide, and the transmembrane (TM) domain [16,19]. Between the S1 and S2 subunits, there is a polybasic PRRAR furin-like cleavage site which is unique to the S protein of SARS-CoV-2 and may, together with the particularly high-binding affinity to the target receptor ACE2 and the peculiarity of a long symptom-free but nevertheless highly infectious time period between infection and appearance of first symptoms or asymptomatic transmission [20], be responsible for the particularly efficient spread of SARS-CoV-2 compared to previous pathogenic hCoVs. The ACE2 receptor is widely expressed in pulmonary and cardiovascular tissues, which may explain the broad range of pulmonary and extrapulmonary effects of SARS-CoV-2 infection on the cardiac system, gastrointestinal organs, and kidneys [21,22,23]. Most individuals infected with SARS-CoV-2 show mild-to-moderate symptoms, and up to 20% of infections may be asymptomatic. This ratio, however, can differ for different virus variants. Symptomatic patients show a wide spectrum of clinical manifestations ranging from mild febrile illness and cough up to acute respiratory distress syndrome (ARDS), multiple organ failure, and death. Whereas for the Omicron variant, a generally lower severity is found compared to the original Wuhan strain and previous VoCs, the clinical picture of severe cases of COVID-19 in general is rather similar to that seen in SARS-CoV-1- and MERS-CoV-infected patients [10]. Whereas younger individuals show predominantly mild-to-moderate clinical symptoms, elderly individuals frequently exhibit severe clinical manifestations [24,25,26,27]. Pre-existing comorbidities, including diabetes, respiratory and cardiovascular diseases, renal failure and sepsis, older age and male sex, are associated with more severe disease and higher mortality [28,29,30,31].

Regarding the pathophysiological manifestations, a diffuse alveolar disease with capillary congestion, cell necrosis, interstitial edema, platelet-fibrin thrombi, and infiltrates of macrophages and lymphocytes are typical for critical and fatal COVID-19 cases [32]. Furthermore, induction of endotheliitis in various organs (including lungs, heart, kidney, and intestine) by the SARS-CoV-2 infection as a direct consequence of viral involvement and of the host inflammatory response has been demonstrated [22,23].

The molecular and cellular mechanisms for the morbidity and mortality of SARS-CoV-2 are getting increasingly clarified. Virus-induced cytopathic effects and viral evasion of the host immune response, and a dysregulated host IFN type I response by SARS-CoV-2 [33], seem to play a role in disease severity. Furthermore, clinical data from patients, in particular those with severe clinical manifestations, show that highly dysregulated exuberant inflammatory and immune responses correlate with the severity of disease and lethality [25,34,35,36]. In particularly, upregulated cytokine and chemokine levels, also termed “cytokine storm”, have been demonstrated to play a central role in the severity and lethality of SARS-CoV-2 infections. Elevated plasma levels of IL-1β, IL-7, IL-8, IL-9, IL-10, G-CSF, GM-CSF, IFNγ, IP-10, MCP-1, MIP-1α, MIP-1β, PDGF, TNFα, and VEGF have been measured in both ICU (intensive care unit) patients and non-ICU patients, with significantly higher plasma levels of IL-2, IL-7, IL-10, G-CSF, IP-10, MCP-1, MIP-1α, and TNFα found in patients with severe pneumonia developing ARDS and requiring ICU admission and oxygen therapy compared to non-ICU patients showing pneumonia without ADRS [25,37]. Several studies have shown that highly stimulated epithelial-immune cell interactions escalate into exuberant dysregulated inflammatory responses with significantly (topically and systemically) elevated cytokine and chemokine release [38,39].

Regarding the underlying signaling pathways, several reports indicate the NF-κB pathway as one of the critical signaling pathway for the SARS-CoV-2 infection-induced proinflammatory cytokine/chemokine response, playing a central role in the severity and lethality of COVID-19, probably in the context of related pathways such as the IL-6/STAT pathway [40,41,42,43,44,45,46]. Notably, this NF-κB-triggered proinflammatory response in acute COVID-19 is shared with other acute respiratory viral infections caused by highly pathogenic influenza A virus of H1N1 (e.g., Spanish flu) and H5N1 (avian flu origin), as well as SARS-CoV-1 and MERS-CoV [10,45].

Excessive activation of exuberant inflammatory responses with involvement of endothelial cells, epithelial cells, and immune cells may lead to further disturbances of other integrated systems, such as the complement system, coagulation, and bradikinine systems, leading to increased coagulopathies and positive signaling feedback loops accelerating COVID-19-associated inflammatory processes [47,48,49,50,51,52,53]. Furthermore, vascular occlusion by neutrophil extracellular traps (NETs) [54,55] and disturbances of coagulation including thromboses and multiple microthromboses seem to be another (beside cytokine storm) hallmark of the COVID-19 disease. The development of coagulopathies is one of the key features associated with poor outcome, with elevated D-dimer levels, prolonged prothrombin time, thrombocytopenia, and low fibrinogen (indicating fibrinogen consumption) found as prognostic indicators for poor outcome [56,57,58,59]. Lung histopathology often reveals fibrin-based blockages in the small blood vessels of patients who succumb to COVID-19 [32]. Furthermore, various types of antiphospholipid (aPL) antibodies targeting phospholipids and phospholipid-binding proteins were found in half of the serum samples from patients hospitalized with COVID-19. Higher titers of aPL antibodies were associated with neutrophil hyperactivity, including the release of neutrophil extracellular traps (NETs), higher platelet counts, and more severe respiratory disease [60]. High rates of thrombotic-related complications have been reported in adult patients with severe COVID-19 as well as in children developing COVID-19 or multisystem inflammatory syndrome (MIS-C). Studies in adults have invoked thrombotic microangiopathy (TMA) from endothelial cell damage to small blood vessels, leading to hemolytic anemia, thrombocytopenia, and organ damage [61,62,63,64,65].

## 3. Molecular Changes in the Omicron’s Spike Protein and Their Impact on Transmissibility, Immune Escape, and Pathogenicity

Omicron has gathered more than 30 amino acid substitutions, one insertion of three amino acids, and three small deletions in the spike protein compared to the original Wuhan strain, with various mutations shared with previous VoCs and 26 unique modifications. The mutations in the Omicron spike protein can be grouped more or less by four distinct parts of the molecule: the NTD, the RBD, near the S1/S2 cleavage site, and in the S2 subunit with accumulation in the HR1 region [3]. The structure of the spike protein and the mutations found in the Omicron variant are shown in Figure 1:

The highest number of the mutations in the spike protein are in the S1 region, whereas the S2 is relatively conserved in the Omicron variant and actually harbors only six unique mutations i.e., N764K, D796Y, N856K, Q954H, N969K, and L981F, which are not detected in other variants of concern. Remarkable is the increase in positively charged amino acids in four out of these six mutations, plus one change leading to the loss of a negative charge (D796Y). Additionally, mutations in the RBD region led to a significant increase in positively charged residues (N440K, T478K, Q493R, Q498R, Y505H, and T547K), and the S1/S2 cleavage site amino acid substitutions lead to positively charged amino acids (N679K, P681H). In contrast, there is a negative charge accumulation on the NTD surface derived from G142D and EPE insertion after R214 [3].

### 3.1. Mutations in the Omicron Spike Protein RBD Region Strengthen the Spike-ACE2 Interaction

Mutations in the RBD region have been shown to intensify the interface interaction with ACE2 (Figure 2).

Computational mutagenesis and binding free energy analyses confirmed that Omicron spike protein binds ACE2 are stronger than wild-type SARS-CoV-2. Notably, three substitutions to positively charged amino acids in the RBD, i.e., T478K, Q493R, and Q498R, significantly contribute to the binding energies and doubled electrostatic potential of the RBD^Omic-^ACE2 complex, suggesting that the Omicron binds ACE2 with greater affinity, enhancing its infectivity and transmissibility [6]. Other recently published Cryo-EM structural analyses of the Omicron variant spike protein in complex with human ACE2 revealed new salt bridges and hydrogen bonds formed by the mutated residues Q493R, G496S, and Q498R in the RBD with ACE2 [67,68]. Furthermore, the N501Y and S477N (distinct to Omicron) mutations enhance transmission primarily by enhancing binding [69] with the N501Y mutation, which is common to all VoCs except the Delta variant, enhancing binding to ACE2 receptor by a factor of 10 compared to the Wuhan strain spike protein [70].

Apart from the significant increase in binding affinity to ACE2 by amino acid substitutions in the Omicron spike RBD region, the increase in positive charge in various regions of the spike protein may increase also binding to some of the various proposed co-receptors for SARS-CoV-2, in particular those with high negative charge such as heparane sulfate and sialic acid [11].

### 3.2. Enhanced Escape from Therapeutic Antibodies and Immune Sera by Mutations in RBD and NTD

Mutations in the NTD and RBD region have been shown to affect binding of therapeutic monoclonal antibodies and immune sera from vaccinated individuals. Instead of E484K substitution that helped the neutralization escape of Beta, Gamma, and Mu variants, the Omicron variant harbors the E484A substitution. Together, T478K, Q493R, Q498R, and E484A substitutions contribute to a significant drop in the electrostatic potential energies between RBD^Omic-^mAbs, in six out of seven tested therapeutic antibodies: Etesevimab (AbCellera&Eli Lilly), Bamlanivimab (AbCellera&Eli Lilly), CTp59 (Celltrion), Imdevimab/REGN10987 (Regeneron), Casirivimab/REGN10933) (Regeneron), and a moderate drop for AZD9995 (Astrazeneca). Regarding the question of which mutations are particularly involved in weakening the RBD^Omic-^mAb interactions, calculated changes in energy indicated that, e.g., for Bamlanivimab and CT-p59, highly stable salt bridges were lost due to E484A mutation or due to the combination of E484A, Q493K, and Y505H, respectively, in RBD^Omic^. These data suggest that mutations in the Omicron spike were precisely selected to utilize the same mutations to enhance receptor binding and resist antibody binding [6]. Another study provided further evidence that amino acid substitutions in the RBD, E484A, and Q493R impact interactions with Casivirimab (REGN 10933) and S375F and N501Y with Imdevimab (REGN 10897). Indeed, whereas the antibodies individually were partially effective and inhibited highly potent against VoC Delta in combination, there was a complete loss of neutralizing activity against Omicron [3]. Furthermore, testing the neutralizing activity of sera from serum samples derived from persons vaccinated two times with either BNT162b2, mRNA 1273, ChAdOx-1, or Coronovac showed more than a 10-fold decrease in neutralization activity for Omicron compared to Delta, with almost no neutralizing activity against Omicron found for ChAdOx-1 in serum samples after two immunizations, and only low activity against Delta and no activity against Omicron found for individuals immunized with Coronavac [3].

The Omicron spike contains some of the mutations also reported in previous VoC, particularly D614G, found in all VoCs, which has been shown to enhance the receptor-binding by increasing its “up/open” conformation necessary for binding of the RBM to ACE2 and to enhance the overall density of the spike protein at the virus’ surface [71,72]. The Omicron unique insertion mutation, i.e., Ins214EPE, maps to the NTD distant from the known antibody binding sites. However, the loop with the insertion maps to known human T-cell epitope on SARS-CoV-2 [73].

For multiple mutations in the RBD and NTD regions of Omicron, the correlation with enhanced escape behavior has been found or suggested [74]. The deletion Δ143–145 is also found in the spike protein of the Alpha variant. The resistance of Alpha to most monoclonal antibodies in the NTD is largely conferred by this deletion. Mutations in this region abolish the binding of monoclonal antibody 4A8 [75,76]. The N440K mutation was observed in a virus isolated in India associated with patient re-infection and described as an “immune escape variant” [77] and emerged under selection pressure against the human monoclonal antibody C135 [78]. For G446S, other mutations at this position in different variants have conferred escape from multiple antibodies [79,80]. The S477N mutation in the spike protein is resistant to neutralization by multiple monoclonal antibodies and resulted in a degree of resistance across the entire panel of antibodies [80]. The T478K mutation is also present in Delta [81]. Other alterations at this position that have provided resistance to neutralizing antibodies confers resistance to monoclonal antibody 2B04 and 1B07 [80]. Q493R mutations escape neutralization by the monoclonal antibody cocktail LY-CoV555 + LY-CoV016 [82] and confers a greater than two log reduction in IC_50_ for the REGN10989/10934 pair of monoclonal antibodies compared to the protein [83]. The Q498R mutation confers escape against the COV2-2499 antibody non-mutated spike [79].

### 3.3. Mutations in S2

The mutations found in the S2 subunit are of particular interest in relation to another essential step in the virus infection process, i.e., the fusion of the virus with the host cell membrane. S2 is a typical viral class I fusion protein, which includes a hydrophobic fusion peptide (FP), two α-helical hydrophobic (heptad) repeats (HR1 and HR2), a long, linking loop region, and a transmembrane domain. HR2 is located close to the transmembrane anchor, and HR1 is close to the FP. Binding of the S1 receptor binding domain to the ACE2 receptor on the target cells triggers a series of conformational changes in the S2 subunit, resulting in the proteolytic cleavage between S1 and S2, and its transition from a prefusion metastable form to a postfusion stable form, with insertion of the putative fusion peptide into the lipid layer of the target cell membrane due to the abundance of hydrophobic residues. The precise localization of the fusion peptide (FP) in the S2 fusion protein in SARS-CoV-2 is still under controversion. For SARS-CoV-2, a stretch around the amino acid sequence _788_IYKTPPIKDFGGFNFSQIL_806_ [16,19] has been suggested to be involved in membrane fusion. The fusion peptide is characterized by its higher hydrophobicity, due to a high density of nonpolar amino acid residues, such as glycine (G), alanine (A), phenylalanine (F), leucine (L), Isoleucine (I), proline (P), and tyrosine (Y). This hydrophobic core plays an essential role in the interaction and penetration into the host membrane lipid. A computer-aided drug design study using FDA-approved small molecules docking to the fusion peptide hydrophobic pocket of S2 suggested that the potential binding site at the fusion peptide region is centralized amid the Lys790, Thr791, Lys795, and Asp808 residues (with some additional interactions also near Gln872) [84]. This is followed by a further conformational change, leading to the association of the two heptad repeat (HR) regions HR1 and HR2 domains to form a six-helix bundle fusion core structure (6HB) motif where the HR1 helices form a central coiled–coil fusion core surrounded by three HR2 helices in an anti-parallel arrangement (see Figure 3C) [85]. This brings the viral envelope and target cell membrane into close proximity enabling fusion. The six-helix bundle is linked by a beta-hairpin loop, which finally acts as a hinge end-to-end in-groove attachment of HR1 and HR2 [84]. Notably, the formation of 6HB in class I fusion proteins is a common step in viral entry and is used by various virus types, including HIV-1, Parainfluenza, Newcastle disease, Respiratory syncytial virus, Herpes simplex virus, Ebola virus, as well as members of the coronaviridae family, including HCoV-229E, MERS-CoV, and SARS-CoV-2. For insertion into the cellular membrane, the FP must be accessible, which is achieved through cleavage at the S2′ site by the transmembrane serine protease 2 (TMPRSS2). Alternatively, the pH-dependent enzyme cathepsin L can take over the processing function after endocytic uptake [86]. Notably, endocytic uptake and activating cleavage by cathepsin L may also overcome the requirement for furin-mediated priming of the S protein [14,87,88].

Regarding the Omicron-unique mutations in the S2 subunit, most of them have been suggested primarily to stabilize the spike trimer. The D796Y mutation replaces a charged surface-exposed acidic residue with tyrosine, containing an aromatic side chain allowing for potential carbohydrate-pi interactions with the N-linked glycan chain originating from N709 of the neighboring monomer chain, this way having a stabilizing effect for the spike trimer. In addition, for the N856K mutation, the longer side-chain of the lysine residue has been suggested to form new interactions with T572 from an adjacent monomer. For N764K, two new interactions of an amine head-group of K764 with Q314 and N317 from an adjacent monomer, are expected to stabilize the Spike trimer [89].

We used the PDB 7TEI and PDB 7T9K Cryo-EM structures of the Omicron spike protein file [66,67] to visualize the amino acid substitutions in the S2 subunit and in the interface area between S1 and S2. The substitutions Q954H, N969K, and L981F are located within the HR1 region and D796Y is located in the area of a putative fusion protein region. Notably, together with the spatially and proximally located N764K and N856K, these mutations represent four changes to positively charged amino acids and one exchange of the negatively charged glutamic acid by a neutral tyrosine (D796Y), with an increase at five sites of the positive electrostatic charge in the S2 region (see Figure 3A,B), which may have an impact on interaction with various innate receptors, as discussed below.

### 3.4. Mutations near to the S1/S2 Furin-Like Cleavage Site

The _861_PRRAR_865_ polybasic furin-like S1/S2 cleavage site plays a central role in the highly effective plasma membrane route of viral entry, being a necessary precedent cleavage step for the following S2′ cleavage by the cellular serine protease TMPRSS2 after binding of S1 to ACE2, provided that the cells express both ACE2 and TMRPSS2. S1/S2 cleavage at the polybasic furin-like cleavage site occurs primarily during virion release from the producer cells, but secreted furin may also enable S1/S2 cleavage of the spike protein outside the cells [90,91]. In contrast, the endosomal entry route used in cells not expressing ACE2 and/or TMPRSS2 does not require spike cleavage at the furin cleavage site but was described to be approximately 100–1000× fold less effective [13,90,92] due to restricting factors in the endosomes and enhanced recognition by the host innate immune receptors resulting in activation of antiviral cellular pathways. The highly efficient plasma membrane entry route with furin and TMPRSS2 cleavage correlates with enhanced cell fusion, leading to the formation of syncytia between multiple virus producer cells which is expected to significantly enhance viral production and pathogenicity of the virus [90,91,92,93,94,95]. Using a reverse genetic system, a SARS-CoV-2 mutant that lacked the furin cleavage site (ΔPRRA) in the S protein was generated. The deletion of PRRA reduced S protein cleavage but augmented viral replication in Vero E6 cells, which are deficient in TMPRRS2. Ectopic expression of TMPRSS2 in Vero E6 cells removed the fitness advantage for ΔPRRA SARS-CoV-2. By contrast, the ΔPRRA mutant was attenuated in a human respiratory cell line and had reduced viral pathogenesis in both hamsters and K18-hACE2 transgenic mice (which express human ACE2) [96].

The ACE2/TMPRSS2 pathway is the preferable pathway for SARS-CoV-2 to enter lung cells, such as alveolar AT1 and AT2 pneumocytes, whereas upper airway cells (expressing significantly lower amounts of ACE2 and/or TMPRSS2) seem to employ preferably or exclusively the endosomal entry route [3]. Structurally, the loop containing the S1/S2 cleavage site is largely flexible and extends outwards, exposing the cleavage site for furin in both the Delta and Omicron models. Omicron has three amino acid substitutions near the S1/S2 cleavage site: H655Y, N679K, and P681H. The substitution N679K is distinct to Omicron with no effects on the S1/S2 cleavage described so far. The H655Y substitution is also found in the Gamma variant, and substitutions at the P681 position are found in various previous VoC, i.e., P681R in the case of Delta and P681H in the Alpha variant, similar to Omicron. Substitution of the P681 (in the original Wuhan variant) by positively charged amino acids such as Arg in the case of Delta has been shown to significantly increase spike protein cleavage, enhanced syncytia formation leading to higher viral transmission, and higher pathogenicity compared to the D614G Wuhan-1 spike [94,97,98].

Furthermore, P681R was also shown to stimulate NF-κB and AP-1 signaling in human monocytic THP1 cells and to induce significantly higher levels of pro-inflammatory cytokines [98]. Different to P681R, as present in Delta, Omicron has the P681H substitution also found in the Alpha variant. Studies on the effect of the P681H substitution in the Alpha variant showed a moderate tendency for an increase in its cleavability by furin-like proteases, but that did not translate into increased virus entry or membrane fusion, which were roughly equal to the Wuhan wild-type [98]. On the other hand, improved viral fitness was suggested for both P681H and P681R SARS-CoV-2 Gamma variants [99]. Accordingly, the P681H in Omicron could be expected to have a similar effect as in the Alpha variant, with no major effect on transmissibility, and at least no negative impact.

Surprisingly, the Omicron variant was found to have significantly decreased S1/S2 cleavage, significantly lower infectivity in TMPRRS2-rich Calu-3 cells, and equal or higher infectivity to TMPRRS2-deficient H1299 cells, compared to wild-type or Delta variants, as well as almost completely absent cell fusion, compared to pronounced cell fusion and syncytia formation after the wild-type or Delta infection. Omicron spike pseudotyped virus (PV) entry into lower airway organoids and Calu-3 lung cells was impaired. In lung cells expressing TMPRSS2, the Omicron virus showed significantly lower replication in comparison to Delta. Cell–cell fusion mediated by spike glycoprotein is known to require S1/S2 cleavage and the presence of TMPRSS2. Fusogenicity of the Omicron BA.1 spike was severely impaired despite TMPRSS2 expression, leading to marked reduction in syncytium formation compared to Delta spike. These data indicate that suboptimal Omicron S1/S2 cleavage reduces efficient infection of lower airway cells expressing TMPRSS2, but not in TMPRSS2 negative cells, such as those found in the upper airways [3]. These results were rather unexpected from the molecular modelling perspective. Whereas the lower S1/S2 cleavage in Omicron compared to Delta (P681R) can be explained by the different substitutions for Omicron (similar as Alpha) P681H, the significantly lower S1/S2 cleavage compared to wild-type cannot be explained by this substitution. Moreover, if a difference was expected, then a slightly increased cleavage would have to be expected. On the other hand, these data correlate with recent preclinical and clinical data for Omicron [7,8].

## 4. Lower Pathogenicity of Omicron Compared to Previous VoCs

There are accumulating data that Omicron, despite a significantly higher transmissibility and infectivity, shows lower numbers of severe clinical courses compared to previous VoCs, in particular compared to the Delta variant [8]. Whereas a lower number of severe clinical outcomes in Africa could also be a result of the younger average age in the African population or other continent specific factors [100], the early reports about lower clinical severity of the SARS-CoV-2 Omicron variant in South Africa are meanwhile supported by concordant reports from other geographical regions of the Omicron pandemic (e.g., UK and USA), showing high transmissibility but significantly lower pathogenicity [101,102].

A retrospective cohort study of electronic health record (EHR) data of 577,938 first-time SARS-CoV-2 infected patients from a multicenter, nationwide database in the US during 1 September 2021–24 December 2021, including 14,054 who had their first infection during the 15 December 2021–24 December 2021 period, when the Omicron variant emerged (“**Emergent Omicron cohort**”) and 563,884 who had their first infection during the 1 September 2021–15 December 2021 period when the Delta variant was predominant (“**Delta cohort**”) was conducted. The 3-day risks of four outcomes (ED visit, hospitalization, ICU admission, and mechanical ventilation) were compared. The 3-day risks in the Emergent Omicron cohort outcomes were consistently less than half those in the Delta cohort for all parameters tested: ED visit: 4.55% vs. 15.22% (risk ratio or RR: 0.30, 95% CI: 0.28–0.33); hospitalization: 1.75% vs. 3.95% (RR: 0.44, 95% CI: 0.38–0.52]); ICU admission: 0.26% vs. 0.78% (RR: 0.33, 95% CI:0.23–0.48); mechanical ventilation: 0.07% vs. 0.43% (RR: 0.16, 95% CI: 0.08–0.32). In children under 5 years old, the overall risks of ED visits and hospitalization in the Emergent Omicron cohort were 3.89% and 0.96%, respectively, significantly lower than 21.01% and 2.65% in the matched Delta cohort (RR for ED visit: 0.19, 95% CI: 0.14–0.25; RR for hospitalization: 0.36, 95% CI: 0.19–0.68). Similar trends were observed for other pediatric age groups (5–11, 12–17 years), adults (18–64 years), and older adults (≥65 years). In summary, the data indicate that first time SARS-CoV-2 infections occurring at a time when the Omicron variant was rapidly spreading were associated with significantly less severe outcomes than first-time infections when the Delta variant predominated [102].

This clinical picture of attenuated severity is also supported by animal data showing high, TMPRSS2-independent, replication in the upper airway organs, but lower pathogenicity in animal studies. The ability of multiple B.1.1.529 Omicron isolates to cause infection and disease in immunocompetent and human ACE2 (hACE2)-expressing mice and hamsters was studied. Despite modeling and binding data suggesting that the B.1.1.529 spike can bind more avidly to murine ACE2, the authors observed attenuation of infection in three different mouse models, i.e., 129, C57BL/6, and BALB/c mice, as compared with previous SARS-CoV-2 variants, with limited weight loss and lower viral burden in the upper and lower respiratory tracts [7]. Although K18-hACE2 transgenic mice sustained an infection in the lungs, these animals did not lose weight. In wild-type and hACE2 transgenic hamsters, lung infection, clinical disease, and pathology with Omicron also were milder compared to historical isolates or other VoCs. Overall, these studies using several different Omicron isolates demonstrated attenuated lung disease in rodents, which parallels preliminary human clinical data [7,8,9]. A lower virus replication in lungs, together with a faster replication of the upper respiratory system, such as nasopharyngeal and bronchi [9], could explain to a large extent Omicron’s greater ability for transmission between people while apparently causing less severe COVID-19 disease. However, the molecular mechanisms responsible for this reciprocal change in tropism to upper vs. lower respiratory system in the Omicron variant are so far not defined. The answer to this question may be hidden in the involved signaling pathways and the characteristics and the distribution of the triggering receptors, as discussed below.

### 4.1. NF-κB Pathway Activation by SARS-CoV-2

There are increasing data for the central role of the NF-κB signaling pathways for the SARS-CoV-2 infection-induced proinflammatory cytokine/chemokine response, and severity and lethality of COVID-19. This exaggerated NF-κB-triggered proinflammatory response in acute COVID-19 is shared also with other acute respiratory viral infections caused by the highly pathogenic influenza A virus of H1N1 (e.g., Spanish flu) and H5N1 (avian flu origin), SARS-CoV, and MERS-CoV [40,41,42,43,44,45,46]. As early as one day post-infection with a SARS-CoV-2 infection, pluripotent stem cell-derived human lung alveolar type 2 cells have been shown to start a rapid epithelial-intrinsic inflammatory response with transcriptomic change in infected cells, characterized by a shift to an inflammatory phenotype with upregulation of NF-κB signaling and loss of the mature alveolar program [39]. Furthermore, characterization of bronchoalveolar lavage fluid immune cells from patients with COVID-19, were compared to healthy donors by using single-cell RNA sequencing and demonstrated proinflammatory monocyte-derived macrophages with an M1 profile with enhanced expression of NF-κB and STAT1/1, accompanied by high cytokine and chemokine expression [103].

### 4.2. SARS-CoV-2 Spike Protein Induces NF-κB

Several studies have studied which part(s) of SARS-CoV-2 are responsible for the massive NF-κB pathway activation. *Khan* et al., showed that the spike (S) protein potently induces inflammatory cytokines and chemokines, including IL-6, IL-1β, TNFα, CXCL1, CXCL2, and CCL2, but not IFNs in human and mouse macrophages. No inflammatory response was observed in response to membrane (M), envelope (E), or nucleocapsid (N) proteins. When stimulated with extracellular spike protein, A549 human lung epithelial cells also produced inflammatory cytokines and chemokines. The spike protein was shown to trigger inflammation via activation of the NF-κB pathway in a MyD88-dependent manner. Both S1 and S2 triggered NF-κB activation, with S2 showing higher potency on an equimolar basis [104].

In a second study upregulation of TLR4, IL1R, NF-κB signaling pathway molecules in COVID-19 patients were found, associated with the altered immune responses to viral components, host damage-associated molecular pattern (DAMP) signals, and cytokine signaling activation, resembling those seen with bacterial sepsis. When testing for different components of SARS-CoV-2, the nucleocapsid (NC) and the S2 subunit of spike proteins were found to activate TLR4 and NF-κB pathways with an expression of multiple pro-inflammatory cytokines and chemokines [105].

In a third study, the spike protein was demonstrated to promote an angiotensin II type 1 receptor (AT1)-mediated signaling cascade, and to activate NF-κB and AP-1/c-Fos via MAPK activation, and IL-6 release [106]. A fourth study demonstrated that the SARS-CoV-2 spike protein S1 subunit induces high levels of NF-κB activation, production of proinflammatory cytokines, and epithelial damage in human bronchial epithelial cells. NF-κB activation required S1 interaction with the human ACE2 receptor and early activation of endoplasmic reticulum (ER) stress and associated un-folded protein response and MAP kinase signaling pathways [107].

In another study, human peripheral blood mononuclear cells (PBMCs) showed significant release of TNFα, IL-6, IL-1β, and IL-8 following stimulation with spike S1 protein. Activation of the NF-κB pathway was demonstrated by phosphorylation of NF-κB p65, IκBα degradation, and increased DNA binding of NF-κB p65 after stimulation with spike S1 protein. NF-κB activation and cytokine release were blocked by treatment with dexamethasone or the specific NF-κB inhibitor BAY11-7082 [108].

Furthermore, SARS-CoV-2 S protein was suggested to bind to LPS. Spike protein, when combined with low levels of LPS, boosted NF-κB activation in monocytic THP-1 cells and cytokine responses in human blood and PBMC, respectively. The study demonstrated that the S protein modulated the aggregation state of LPS, providing a potential molecular link between excessive inflammation during infection with SARS-CoV-2 and comorbidities involving increased levels of bacterial endotoxins [109].

In a mouse model, the S1 subunit of the spike protein was demonstrated to elicit strong pulmonary and systemic inflammatory responses in transgenic K18-hACE2 mice after intratracheal installation, accompanied by loss in body weight, increased white blood cell count, and protein concentration in bronchoalveolar lavage fluid (BALF), and upregulation of multiple inflammatory cytokines by activation of NF-κB and from the signal transducer and activator of transcription 3 (STAT3) [110].

Similar to SARS-CoV-2, the clinical picture of severe acute respiratory syndrome (SARS) is characterized by an overexuberant immune response with lung lymphomono-nuclear cell infiltration that may account for tissue damage more than the direct effect of viral replication. In addition, SARS-CoV purified recombinant S protein was shown to stimulate murine macrophages to produce proinflammatory cytokines (IL-6 and TNFα) and the chemokine IL-8, which were dependent on NF-κB activation [111]. Overall, these data demonstrate that the spike protein of both SARS-CoV-1 and SARS-CoV-2 S induces powerful NF-κB activation, showing strong similarity to data recorded for the SARS-CoV S protein.

### 4.3. NF-κB Is Essential for SARS-CoV-2 Replication

In addition to the multiple lines of evidence showing the critical role of the NF-κB signaling pathway in cytokine/chemokine release and hyper-immune activation, there is an additional set of data indicating that NF-κB is essential for viral replication of SARS-CoV-2 in the host cell. Epigenetic and single-cell transcriptomic analyses showed an early NF-κB transcriptional signature comprised of chemokines (e.g., CXCL8, CXCL10, CXCL11, and CCL20) and proinflammatory cytokines (e.g., IL-1A and IL-6) and upregulated NFKB1A, phospohorylation of IκBa, and NF-κB p65 in a relative absence of an ISG response. There was significantly enhanced enrichment of the NF-κB–related DNA-binding motifs and corresponding increase in genomic accessibility for REL, NKFB1, and RELA, but not for IRF3 and IRF7. Disruption of NF-κB signaling through the silencing of the NF-κB transcription factor p65 or p50 resulted in loss of virus replication that was rescued upon reconstitution. Furthermore, A549-ACE2 cells pre-treated with BAY11-7082 (an inhibitor of IκBα phosphorylation), MG115 (a proteasome inhibitor preventing proteolytic degradation of IκBα), prior to infection with SARS-CoV-2, showed significant inhibition of viral replication following BAY11-7082 treatment and an almost complete loss of viral protein and RNA expression in response to MG115. In addition, a significant reduction of secreted proinflammatory cytokines and chemokines in response to SARS-CoV-2 infection in A549-ACE2 cells after BAY11-7082 was found [112].

There is an analogy with IAV where NF-κB pathway was also found to support IAV infection by enhancing caspase-mediated nuclear export of viral ribonucleoproteins [113,114].

These data suggest that SARS-CoV-2 triggers both hyper-immune activation and viral replication via activation of the NF-κB signaling pathway.

### 4.4. NF-κB, Cytokines, and Hypoxia Enhance Furin Expression

As well as its central involvement in immune hyper-activation and SARS-CoV-2 replication, the NF-κB signaling pathway may also be involved in the modulation of the SARS-CoV-2 host cell type tropism by modulation of the furin-mediated cleavage of the spike protein, with respect to the availability of sufficient protease in the virus producer cells. Within the SARS-CoV-2 replication cycle, the cleavage at the furin site between S1 and S2 most likely occurs during virion assembly, or just before release. This timing correlates with the virus’ passage through the Golgi or lysosomes. Notably, furin has been found primarily in the trans-Golgi-network (TGN)—a late Golgi structure that is responsible for sorting secretory pathway proteins to their final destinations. From the TGN, furin follows trafficking through several TGN/endosomal compartments to the cell surface. The proteolytic activity of furin shows a broad pH optimum, with high enzymatic activity between (pH 5–8)—a pH range covering both TGN and lysosomes [115,116].

By cutting the bond between the S1 and S2 subunits, the furin cut triggers conformational changes in the virion spike protein so that at binding to the next host cell it is accessible to the second cut by TMPRSS2, which exposes the hydrophobic area that introduces into the host cell membrane. If spike protein is not clipped by furin, it bypasses the TMPRRS2 cleavage, and the virus can enter only via the slower and less-efficient endosomal pathway, which results in lower transmissibility to TMPRSS2-dependent cells, such as lung cells, in contrast to TMPRRS2 non-dependent cells, such as cells of the upper airway tissues [117]. Therefore, this process of furin cleavage at the S1/S2 site actually depends on two factors, i.e., the presence of a suitable cleavage site and the availability of the protease. Although furin is rather ubiquitously expressed across most tissues, it is usually expressed at very low levels [90,118], with the exception of few cell types in the brain, salivary gland, pancreas, kidney, and placenta [119].

Whereas furin is expressed usually at very low basic levels, it can be induced in response to hypoxia and cytokine stimulation. Furin is induced by IL-12 in T cells [120,121], with the IL-12 expression depending on NF-κB pathway signaling [122,123]. In contrast to the very low expression level of furin in most normal cells, elevated levels are found in many cancer cells, where furin seems to be closely related to tumor formation and migration [124]. p38 activation in cervical cancer cells was shown to induce NF-κB-dependent expressions of furin. Furin expression and cell motility was impeded by blockades to MKK3/6, p38α/β, or NF-κB signaling [125]. Another study correlated the osteopontin-p38 MAPKinase–NF-κB-furin expression with diabetes mellitus progression and increased risk of diabetes-linked premature mortality and a more severe clinical picture in diabetic patients after SARS-CoV-2 infection [126].

In this context, the role of Hypoxia-induced Factor-1 alpha (HIF-1α) and its connection to NF-κB pathway and furin expression may be important. Hypoxia was shown to stimulate furin expression, with direct HIF-1α action on the furin promoter as a canonical hypoxia-responsive element site with enhancer capability [127]. Regarding the initial signal triggering, this NF-κB-HIF-1α-Furin expression axis, there are various studies demonstrating the cross-talk between Toll-like receptor/NF-κB pathway activation and HIF-1α. The HIF-1α promotor was shown to contain an active NF-κB binding site, upstream of the transcription start site [128]. Extensive cross-talk between hypoxia and inflammation signaling have been described, showing that activation of TLR3 and TLR4 stimulated the expression of HIF-1α through NF-κB [129]. Peroxiredoxin (Prx1), a TLR4 agonist, was shown to stimulate increased NF-κB interaction with the HIF-1α promoter, leading to enhanced promoter activity and increase in HIF-1α mRNA levels, and augmented HIF-1 activity. In turn, Prx1-induced HIF-1α also promoted NF-κB activity, suggesting the presence of a positive feedback loop [130].

Furthermore, agonists of TLR4 (e.g., LPS) and TLR2 (e.g., lipoteichoic acid) have been demonstrated to induce in a NF-κB dependent way the expression of HIF-1α in human monocyte-derived dendritic cells under normoxic conditions [131]. Furthermore, activation of TLR4 by LPS was demonstrated to raise the levels of HIF-1α in macrophages. HIF-1α was shown to be a critical determinant of sepsis promoting the production of inflammatory cytokines, including TNF-α, IL-1, IL-4, IL-6, and IL-12, which reach harmful levels during early sepsis [132]. This is in line with data showing the critical role of TLR4 and NF-κB activation in HIF-1α activation during trauma/hemorrhagic shock-induced acute lung injury after lymph infusion in wild-type mice, in comparison to mice that harbor a TLR4 mutation and/or NF-κB inhibitors [133]. TLR4 was demonstrated to also promote HIF-1α activity by triggering reactive oxygen species in cervical cancer cells by mechanisms involving activation of lipid rafts/NADPH oxidase signaling [134].

Considering NF-κB or HIF-1α-induced furin expression, these studies suggest that NF-κB activation following TLR signaling induced during SARS-CoV-2 infection may be essential for at least three mechanisms: (i) increasing viral replication, (ii) immune hyper activation, and (iii) modulation of cell tropism via alteration of proteases expression such as furin, necessary for the TMPRRS-dependent cellular entry pathway of SARS-CoV-2.

### 4.5. Acute Viral Infections Such as Highly Pathogenic IAV and SARS-CoV-2 Depend on Furin and Stimulate Furin Expression

The correlation between furin cleavage and pathogenicity is not only seen for the *coronaviridae* family but is found much broader among different virus types, including dsDNA viruses (*Herpesviridae* e.g., Human cytomegalovirus), *Papillomaviridae*), (+) ssRNA viruses (including *Coronaviridae, Flaviviridae* (e.g., Yelow fever virus, Dengue virus type 2), (−)ssRNA viruses (including *Filoviridae* (e.g., Ebola virus, Marburg virus), *Orthomyxoviridae* (e.g., avian IAV H5, H7, H9), *Paramyxoviridae* (e.g., Mumps virus, Measles virus, Respiratory syncytial virus), ss-RNA-RT viruses *Retroviridae* (e.g., HIV, Feline foamy virus), and dsDNA-RT viruses *Hepadnoviridae* (e.g., Hepatitis B virus). Cleavage of the furin-like site is dependent on the actual amino acid sequence of the polybasic furin cleavage motif, with the prototypical motif for furin (i.e., PCSK3, PACE = paired basic amino acid cleaving enzyme)) being R-X-K/R-R↓, and for other proprotein convertases PCSK1-7 being K/R-X_n_-K/R↓ (n = 0-,2-,4-, or 6-amino acid spacer). Accordingly, different PCSKs can be involved in cleavage, and are dependent on the actual amino acid sequence around the cleavage site and availability of PCSKs. A well-known example is the haemagglutinin (HA) of different Influenza A viruses (IAV). Generally, HA of mammalian and low pathogenic avian IAV cannot be cleaved by furin (or other PCSKs), as they usually only harbour a mono- or dibasic-cleavage site. Instead, they depend on trypsin-like proteases such as human airway trypsin-like protease (HAT). Expression of such trypsin-like proteases is largely restricted to the respiratory and gastrointestinal tract. In contrast, HA of highly pathogenic avian H5 and H7 influenza A viruses can be cleaved by furin or PCSK5, which are present in many cell types. This is because they acquired a polybasic cleavage site upon insertion of additional lysine and/or arginine residues [135]. Notably, a subset of H9N2 low pathogenic avian influenza A virus strains also harbours R-S-K-R↓ or R-S-R-R↓ sites that are not only cleaved by trypsin-like proteases, but also by PCSKs. However, their cleavage is only efficient in the presence of very high amounts of furin or upon mutation of a glycosylation site in HA. Thus, the ability to exploit furin for efficient HA cleavage is not only determined by the presence of a furin consensus target site, but also by the amounts of furin [136]. Interestingly, beside the dependency of highly pathogenic avian IAV replication on furin, infection with IAV was shown to elevate furin expression in the lungs. Wild-type C57BL/6 mice were infected with the mouse-adapted IAV strain A/PuertoRico/8/1934 H1N1 (PR8). Using quantitative RT-PCR, a significant increase in the expression of furin in whole lung tissue of PR8-infected C57BL/6 mice on Day 6 after infection was found, indicating that the IAV-induced inflammation induces furin expression [137].

Notably, SARS-CoV-2 Gene Expression Omnibus records of up-regulated genes suggest that coronavirus infection triggers an increased expression of both ACE2 and FURIN genes 4 days after infection. These findings were corroborated by the increased furin expression documented in the PBMC of patients with severe acute respiratory syndrome. Furthermore, gene expression data identified numerous significantly enriched records of common human disorders manifesting up-regulation of either ACE or FURIN genes correlating with the risk for clinically severe and lethal coronavirus infection [138].

The expression, methylation, mutation rate, and functional enrichment of furin together with the survival rate and COVID-19 outcomes were analyzed in normal and cancer tissues. The furin expression in tumors was significantly increased in several cancer types, such as esophageal carcinoma (ESCA) and testicular germ cell tumors (TGCT). Furin mutations mostly increased expression of ACE2 and TMPRSS2 in various cancers, indicating furin mutations might facilitate COVID-19 cell entry in cancer patients. In addition, high expression of furin was significantly inversely correlated with long overall survival in various cancer types and correlated with increased susceptibility to SARS-CoV-2 and higher severity of COVID-19 symptoms in cancer patients [138].

The relationship between circulating furin levels, disease severity, and inflammation was studied in 52 SARS-CoV-2 patients vs. 36 healthy control participants. The mean furin and IL-6 levels were significantly higher in the peripheral blood of SARS-CoV-2 compared to the controls (*p*  <  0.001). There was a close positive relationship between serum furin and IL-6, and furin and disease severity (r  =  0.793, *p*  <  0001 and r  =  0,533, *p*  <  0.001, respectively) in patients with SARS-CoV-2. These results suggest that furin may contribute to the exacerbation of SARS-CoV-2 infection and increased inflammation and could be used as a predictor of disease severity in COVID-19 patients [139].

Furthermore, higher furin expression was also found in diseases known to predispose a person to severe COVID-19 symptoms, such as severe asthma and in people such as COPD smokers and COPD ex-smokers. ACE2 levels were significantly increased in sputum of severe asthma compared to mild-moderate asthma. Sputum furin levels were significantly related the presence of severe asthma and were strongly associated with neutrophilic inflammation and inflammasome activation, indicating the potential for a greater morbidity and mortality outcome from SARS-CoV-2 infection in neutrophilic severe asthma [140]. Furthermore, ACE2, furin, and TMPRSS2 expression was significantly increased in small airway epithelium (SAE) and type 2 pneumocytes in smokers with COPD (COPD-CS), and ex-smokers with COPD (COPD-ES), compared to the control group that never smoked. (NC) (*p* < 0.001). Importantly, significant changes were observed for tissue co-expression of furin and TMPRSS2 with ACE2 in SAE, type 2 pneumocytes, and alveolar macrophages (AMs). These markers also negatively correlated with lung function parameters. The increased expression of ACE2, TMPRSS2, and furin in COPD patients are detrimental to lung function and indicate that these patients are more susceptible to severe COVID-19 infection. Increased type 2 pneumocytes suggest that these patients are also vulnerable to developing post-COVID-19 interstitial pulmonary fibrosis or fibrosis in general [141].

## 5. SARS-CoV-2 Activates Innate PRRs

The initial signals for these pathways triggered by SARS-CoV-2 can be expected in the danger signaling of the innate immune system. The host innate immune system can recognize pathogen-associated molecular patterns (PAMPs) via pattern recognition receptors (PRRs) during infection to induce inflammatory responses to eliminate pathogens. The PRR families include Toll-like receptors (TLRs), nucleotide-binding oligomerization domain (NOD)-like receptors, retinoic acid-inducible gene-I (RIG-I)-like receptors, C-type lectin receptors, and the absent in melanoma 2 (AIM2)-like receptors. Typical PAMPs are cell wall components of pathogens, such as bacterial lipopolysaccharide (LPS) and lipoproteins, glycans, and conserved proteins such as flagellin or pathogenic nucleic acids, including viral RNA and DNA. PAMPs comprise moieties which are generally conserved among a broader range of pathogenic species but are distinct from host components. Several PRRs have been reported to be involved in sensing β-coronavirus infection, including melanoma differentiation-associated protein 5 (MDA5) [142], TLR7 [143,144], and NLR family pyrin domain containing 3 NLRP3 [145].

In this context, the MyD88 adaptor protein is known to couple multiple upstream sensors (e.g., TLR) with downstream inflammatory signaling pathways such as NF-κB or IFN-induced response factors following β-coronavirus infection [146]. To determine whether MyD88 or another TLR adapter TRIF (TIR-domain-containing adapter-inducing interferon-β) play a role in SARS-CoV-2-induced inflammatory responses and pathogenesis, a publicly available dataset [147] was analyzed for MyD88 and TRIF expression in patients with differing severities of COVID-19 and showed a positive correlation between MyD88 expression and severity of COVID-19, suggesting that MyD88 is associated with COVID-19 pathogenesis in humans. By contrast, TRIF was significantly elevated only in patients with critical COVID-19. MyD88 is a key adapter shared by all TLRs, with the exception of TLR3, which signals exclusively through TRIF, with all other TLRs utilizing MyD88 to trigger inflammatory cytokine production [148]. Parallel to the expression of MyD88, the expression of TLR1, TLR2, TLR4, TLR5, TLR8 and TLR9 was significantly elevated in patients with severe and critical COVID-19. In contrast, expression of TLR3 did not show any correlation with the disease development of COVID-19, and the expression of TLR7 was increased only in patients with moderate COVID-19. Together, these data suggest an association of MyD88 and a panel of TLRs (i.e., TLR1, TLR2, TLR4, TLR5, TLR8, and TLR9) with disease progression in patients with COVID-19 [149].

In this context, it is interesting to take into consideration the expression profile and expression levels of the different TLR on the relevant target cells. From the Human Protein Atlas [119], the expression profiles for relevant receptors on different target cells normalized to nTPM (i.e., Transcripts per million protein coding genes) provide a good overview of the general expression pattern of different molecules on different host cells. In Figure 4, the expression of different TLRs together with other relevant molecules involved in SARS-CoV-2 pathogenesis are depicted. TMPRSS2 is expressed at significant levels by alveolar cells type 1 and 2, but only at minute quantities on macrophages and endothelial cells. Furin and TLR3 are expressed more broadly by many cell types present in the lungs, including alveolar cells, macrophages, and endothelial cells, at low to moderate levels. In contrast, TLR1, 2, 4, 6, and 7 and Cathepsin L (which is essential in the endosomal uptake of SARS-CoV-2) are expressed predominantly on macrophages. Regarding expressions other than macrophages, TLR4 and MD-2 (i.e., the second component of the TLR4-MD-2 complex) are also expressed on endothelial cells, and TLR2 is also expressed on alveolar type 2 cells. Alveolar type 1 cells generally show very low TLR expression. The highest expression levels are found for Cathepsin L, followed by TLR4/MD-2 and TLR2 on macrophages and endothelial cells, whereas ~10fold lower expression levels are seen for TLR1, 6, and 7, which may correlate with the different pathophysiological relevance in COVID-19, as discussed later.

### 5.1. SARS-CoV-2 Envelop E and Spike Protein Activate TLR2 and NF-κB

Zheng et al., have demonstrated for TLR2 and MyD88 expression their correlation with COVID-19 disease severity. TLR2 was shown to recognize the SARS-CoV-2 envelope E protein as its ligand and resulted in the TLR2-dependent cytokine (TNFα, GM-CSF, G-CSF, IL-6) and chemokine (CXCL10, MCP-1) release and lung damage in K18-hACE2 transgenic mice after infection with SARS-CoV-2. Notably, blocking TLR2 signaling in vivo provided protection against the pathogenesis of SARS-CoV-2 infection [149]. Regarding molecular characterization, the E protein was found to interact physically with the TLR2 receptor in a specific and dose-dependent manner. This interaction was able to engage the TLR2 signalling pathway as demonstrated by its capacity to activate the NF-κB transcription factor and to stimulate the production of the CXCL8 inflammatory chemokine in a TLR2-dependent manner. Inhibition of NF-κB led to significant inhibition of CXCL8 production, whereas the blockade of P38 and ERK1/2 MAP kinases resulted only in a partial CXCL8 inhibition [150].

On the other hand, Khan et al., investigated the direct inflammatory functions of major structural proteins of SARS-CoV-2 and showed that the spike protein potently induces inflammatory cytokines and chemokines, including IL-6, IL-1β, TNFα, CXCL1, CXCL2, and CCL2, but not IFNs in human and mouse macrophages. When stimulated with extracellular spike protein, A549 human lung epithelial cells also produced inflammatory cytokines and chemokines. Interestingly, epithelial cells expressing spike protein intracellularly were non-inflammatory but elicited an inflammatory response in co-cultured macrophages. Biochemical studies revealed that the spike protein triggers inflammation via activation of the NF-κB pathway in a MyD88-dependent manner. Furthermore, activation of the NF-κB pathway was abrogated in TLR2-deficient macrophages. Consistently, administration of the spike protein induced IL-6, TNFα, and IL-1β in wild-type but not in TLR2-deficient mice. In this study, both S1 and S2 subunits were demonstrated to show high NF-κB activation, with S2 showing higher potency on an equimolar basis [104].

### 5.2. SARS-CoV-2 Spike Protein Activates TLR4 and NF-κB

The involvement of another TLR, i.e., TLR4 in the pathogenies of COVID-19 has been shown [151]. TLR4 recognizes multiple pathogen-associated molecular patterns (PAMPs) from bacteria, viruses, and other pathogens. In addition, it recognizes certain damage-associated molecular patterns (DAMPs), such as high mobility group box 1 (HMGB1) and heat shock proteins (HSPs) released from dying or lytic cells during host tissue injury or viral infection [152,153]. TLR4 is mainly expressed on immune cells such as macrophages and dendritic cells where it plays a role in the regulation of acute inflammation, but also on some tissue-resident cell populations, for cell defense in case of infection and/or to regulate their fibrotic phenotype in cases of tissue damage [153,154]. The archetypal PAMP agonist for TLR4 is the gram-negative bacterial lipopolysaccharide [155]. Activation of TLR4 by pathogenic components leads to the production of proinflammatory cytokines via the canonical pathway and/or the production of type I interferons and anti-inflammatory cytokines via the alternative pathway. Unlike other TLRs, TLR4 is present at both the cell surface (main site), where it recognizes viral proteins before they enter the cell, and also in endosomes [156].

TLR4 is important in initiating inflammatory responses, and its overstimulation can be detrimental, leading to hyper-inflammation. Dysregulation of TLR4 signalling has been shown to play a role in the initiation and/or progression of various diseases, such as ischaemia-reperfusion injury, atherosclerosis, hypertension, cancer, and neuropsychiatric and neurodegenerative disorders [157,158,159,160]. Moreover, TLR4 is also important in the induction of the host immune response against infectious diseases such as bacterial, fungal and viral infections, and malaria [161].

Recently, there have been several studies pointing to the role of TLR4 in the pathogenesis of COVID-19 [151,162,163,164,165]. Interestingly, in silico studies have indicated that TLR4 has the strongest protein–protein interaction with the spike glycoprotein of SARS-CoV-2 compared to other TLRs [162]. A recently published study demonstrated that the induction of IL1β by SARS-CoV-2 was completely blocked by the TLR4-specific inhibitor Resatorvid. A surface plasmon resonance (SPR) assay showed that SARS-CoV-2 spike trimer directly bound to TLR4 with an affinity of ~300 nM, comparable to many virus-receptor interactions. THP-1 cells were treated with either the spike protein trimer, the N-terminal domain (NTD), or the receptor-binding domain (RBD) of spike protein, respectively. Only the trimeric protein induced IL1β and IL6, which could largely be blocked by the TLR4 inhibitor Resatorvid. Moreover, spike protein was also able to induce IL1β production in the murine macrophage cell line in a TLR4- and MyD88-dependent manner. Consistently, spike protein induced production of IL1β in the primary bone marrow-derived macrophages and peritoneal macrophages from wild-type, but not from TLR4-deficient mice. The NF-κB inhibitor (JSH-23) was able to suppress IL1β induced by spike protein. Collectively, the SARS-CoV-2 spike protein is capable of interacting with and activating TLR4. Furthermore, macrophages from ACE2-deficient or human ACE2-transgenic mice were treated with spike protein. Interestingly, deficiency of ACE2 or overexpression of human ACE2 did not affect the induction of IL1β. Treatment with an ACE2 inhibitor (MLN-4760) or soluble ACE2 was not able to inhibit the induction of IL1β by LPS or spike protein. Moreover, TMPRSS2-specific inhibitor (Bromhexine hydrochloride) did not alter the induction of IL1β by spike protein. Thus, activation of TLR4 by spike protein was not regulated by ACE2 and TMPRSS2 or virus entry. Notably, the induction of IL1β by trimeric spike proteins from SARS-CoV-2 or SARS-CoV was comparable to LPS treatment [163].

In response to exposure to the SARS-CoV-2 spike protein S1, subunit murine peritoneal exudate macrophages produced pro-inflammatory mediators, including TNF-α, IL-6, IL-1β, and nitric oxide. Exposure to S1 also activated NF-κB and c-Jun N-terminal kinase (JNK) signaling pathways. Pro-inflammatory cytokine induction by S1 was suppressed by selective inhibitors of NF-κB (BAY 11-7082) and JNK pathways (SP600125). Treatment of murine peritoneal exudate macrophages and human THP-1 cell-derived macrophages with a TLR4 antagonist attenuated pro-inflammatory cytokine induction and the activation of intracellular signaling by S1 and lipopolysaccharide. Similar results were obtained in experiments using TLR4 siRNA-transfected murine RAW264.7 macrophages. These results suggest that the SARS-CoV-2 spike protein S1 subunit activates TLR4 signaling to induce pro-inflammatory responses in murine and human macrophages [164].

Furthermore, the SARS-CoV-2 spike S1 domain was shown to act as a TLR4 agonist in rat and human cells and to induce a pro-inflammatory M1 macrophage phenotype in human THP-1 monocyte-derived macrophages. Adult rat cardiac tissue resident macrophage-derived fibrocytes (rcTMFs) were treated with either bacterial LPS or recombinant SARS-CoV-2 spike S1 glycoprotein. THP-1 monocytes were differentiated into M1 or M2 macrophages with LPS/IFNγ, S1/IFNγ, or IL-4. TLR4 activation by spike S1 or LPS resulted in the upregulation of ACE2 in rcTMFs. Likewise, spike S1 caused TLR4-mediated induction of the inflammatory and wound-healing marker COX-2 and concomitant downregulation of the fibrosis markers CTGF and Col3a1, similar to LPS. The specific TLR4 inhibitor CLI-095 (Resatorvid^®^), blocked the effects of spike S1 and LPS, confirming the spike S1 subunit as a TLR4 agonist. Confocal immunofluorescence microscopy confirmed 1:1 stoichiometric spike S1 co-localization with TLR4 in rat and human cells. Furthermore, proximity ligation assays confirmed spike S1 and TLR4 binding in human and rat cells. Spike S1/IFN-γ treatment of THP-1-derived macrophages induced pro-inflammatory M_1_ polarization, as shown by an increase in IL-1β and IL-6 mRNA [165].

A model has been proposed in which the SARS-CoV-2 spike glycoprotein binds TLR4 and activates TLR4 signalling, resulting in increased cell surface expression of ACE2 facilitating virus entry. Furthermore, SARS-CoV-2-induced myocarditis and multiple-organ injury may be due to TLR4 activation, aberrant TLR4 signalling, and hyperinflammation in COVID-19 patients. Therefore, TLR4 may be assumed to contribute significantly to the pathogenesis of SARS-CoV-2. TLR4 appears to be a promising therapeutic target in COVID-19, supported by the fact that TLR4 antagonists have been previously used in sepsis and in other antiviral contexts [165].

### 5.3. TLR Activation during Different Highly Pathogenic Viral Infections

There is a growing list of viruses that induce an inflammatory response during acute infection through TLR4 activation. Known TLR4-activating viral proteins include the RSV fusion protein (F), the EBOV glycoprotein, the vesicular stomatitis virus glycoprotein (VSV G), and the dengue virus (DENV) nonstructural protein 1 (NS1). Notably, all infections by these viruses are also characterized by excessive inflammatory responses, which are characterized by elevated levels of a broad array of pro-inflammatory cytokines and chemokines and are associated with serious morbidity and mortality. Examples include acute lung injury caused by infections with respiratory syncytial virus (RSV), highly pathogenic IAV, or SARS-CoV. Excessive inflammatory responses induced by viral infections are not restricted to the lung but can be systemic, as found for Ebola virus (EBOV) disease and severe dengue fever [166,167,168,169,170,171] The Ebola virus glycoprotein was demonstrated to activate the innate immune response in vivo via TLR4 activation, accompanied by multiple cytokine and chemokine expression, which could be inhibited by TLR4 antagonists [172] and was accompanied by pronounced NF-κB activation [173]. There are several commonalities between these viral TLR4 activators. These proteins are all membrane-associated. VSV G, RSV F, and EBOV glycoprotein as well as the spike proteins of SARS-CoV and SARS-CoV-2 are classical viral glycoproteins that are exposed on the surface of viral particles and mediate fusion with host cell membranes through a hydrophobic fusion peptide. The fusion domain is only exposed after conformational changes that occur at the plasma membrane (RSV F, SARS-CoV, SARS-CoV-2) or in the endosome (VSV G, EBOV glycoprotein) [174]. DENV NS1 exists in multiple forms, including a secreted, membrane-bound form [175,176]. The hydrophobic fusion peptide in the RSV fusion protein has been suggested to bind into the deep hydrophobic pocket of MD-2, similar to LPS, to mediate TLR4 activation [171]. TLR4 is stimulated by membrane-bound EBOV glycoprotein and a secreted, cleaved form (shed glycoprotein), both of which retain the hydrophobic fusion domain, but not by a different secreted version of EBOV glycoprotein, i.e., soluble glycoprotein, which lacks the fusion peptide [177,178]. In addition, although DENV NS1 lacks a fusion peptide, it contains exposed hydrophobic domains that mediate membrane interaction and could play a role in TLR4 activation [176].

TLR4 antagonists which suppress LPS-induced TLR4 signaling through competitive interaction with MD-2, such as LPS from the bacterium Rhodobacter sphaeroides (LPS-RS) and Eritoran, also suppress RSV F-, EBOV glycoprotein-, and DENV NS1-mediated TLR4 activation [171,172,175,179,180,181], suggesting a similar mechanism of action. It remains to be determined how each of these glycoproteins interact with the TLR4 receptor complex and in what way the hydrophobic regions are made accessible for interaction with MD-2 and TLR4 leading to dimerization of the TLR4-MD-2 complex (see below). VSV G, RSV F, EBOV glycoprotein, and DENV NS1 are all glycosylated. So far, glycosylation of EBOV glycoprotein seems to be required for TLR4 activation, but it is not known whether this is also the case for the other viral glycoproteins [182].

### 5.4. Activation of TLR4 by LPS

TLR4, which is mainly expressed on cells of the innate immune system, including monocytes, macrophages, and dendritic cells, has long been recognized as the PRR that senses lipopolysaccharide (LPS), a component of the outer membrane of gram-negative bacteria, which can be regarded as the archetypal PAMP agonist for TLR4 [154]. Accordingly, activation of TLR4 by LPS has been studied in great detail. During the initial step, the LPS binding protein (LBP) extracts LPS from bacterial membranes and transfers it to the TLR4 co-receptor cluster of differentiation 14 (CD14). CD14 breaks down LPS aggregates and transfers monomeric LPS into a hydrophobic pocket on myeloid differentiation factor 2 (MD-2, Lymphocyte antigen 96, Ly96), which is part of the MD-2/TLR4 complex. The high-affinity binding of LPS leads to dimerization and activation of the TLR4-MD-2 complex. Dimerization of the TLR4-MD-2 complex results in the recruitment of the intracellular adaptor protein MyD88. The MyD88 aggregation signal is transmitted to IL-1 receptor kinase (IRAK) through an interaction between the death domain of MyD88 and IRAK. Phosphorylation of the signaling kinases eventually activates the transcription factors, NF-κB and activator protein 1 (AP-1) via a signaling cascade, ultimately resulting in the expression and secretion of pro-inflammatory mediators. Apart from the MyD88-dependent pathway, TLR4 dimerization can also activate the TRIF (TIR-domain containing adaptor inducing interferon-β) pathway, which activates interferon response factors to produce and secrete type-I interferons [182].

The archetypal TLR4 agonist LPS is a macromolecular glycolipid composed of the hydrophobic Lipid A attached to a long and branched carbohydrate chain. The Lipid A portion, which is responsible for most of the immunologic activity of LPS, is composed of a phosphorylated diglucosamine backbone with four to seven acyl chains attached to it. The carbohydrate region of LPS comprises the core and the O-specific chain composed of multiple carbohydrate repeating units. Removal of the entire carbohydrate chain by acid hydrolysis has only a minimal effect on the inflammatory activity of LPS, demonstrating only a minor role in recognition by host immune receptors [183].

The extracellular domain of TLR4, TLR1, TLR2, and TLR6 belongs to the Leucine-Rich Repeat proteins (LRR proteins) and is responsible for ligand binding and receptor dimerization. The structure of TLR4 is defined by the conformation of the LRR motifs. Its N-terminal and central domains provide charge complementarity for binding of its surface to its co-receptor MD-2, forming a stable heterodimer. MD-2 is smaller than TLR4 and is the main LPS binding module of the TLR4/MD-2 receptor complex. MD-2 has a β-cup fold structure, composed of two antiparallel β sheets. The two sheets are separated from each other so that the hydrophobic interior is accessible for interaction with ligands. This large internal pocket is ideally shaped for binding flat hydrophobic ligands such as LPS. MD-2 binds to TLR4 primarily via hydrogen bonds and charge interactions, and a few hydrophobic residues in the binding interface.

The crystal structure of the TLR4/MD-2 complex bound to E. coli LPS has been determined and is available at RCSB PDB 3VQ2 [184]. LPS binding induces the formation of the ‘M’ shaped 2:2:2 TLR4/MD-2 and LPS complex (Figure 5). The acyl chains of Lipid A are inserted into the MD-2 pocket and the two phosphate groups of Lipid A form charge and hydrogen bond interactions with charged and polar amino acid residues of the two TLR4 molecules. LPS binding causes dimerization of the TLR4/MD-2 complexes because Lipid A creates an additional binding interface between TLR4(II) and MD-2(I) coming from the two preexisting TLR4/MD-2 (see Figure 5) complexes, respectively. The dimerization interface of MD-2 interacts with a convex surface provided by a small hydrophobic patch in the C-terminal domain of the second TLR4 (Figure 5, TLR4 (II), brown). This dimerization is supported by the interaction between Lipid A inserted in the hydrophobic pocket of MD-2(I) with TLR4(II) (Figure 5). TLR4 agonists are presumable all interacting at this dimerization interface actually enabling dimerization.

The structure–activity relationships of LPS have been studied using natural and chemically modified LPS. The crystal structure of TLRD/MD-2–LPS provides an explanation as to why LPS with six lipid chains is optimal for activation of TLR4 signaling. In the crystal structure, five of the six lipid chains of *E. coli* LPS are completely buried inside the pocket, but the remaining chain is partially exposed to the MD-2 surface and forms the hydrophobic interaction interface together with hydrophobic surface residues of TLR4(II) [183]. The two phosphate groups attached to the glucosamine of LPS seem to be essential for the formation of the stable TLRD/MD-2 complex by making charge- and hydrogen- bond interactions simultaneously with the two TLRs in the complex. Removal of the phosphate groups dramatically reduces or even completely abolishes the inflammatory activity of LPS. Studies on the interdependence of molecular charge and conformation of natural and chemically modified LPS or Lipid A and IL-6 production after stimulation of whole blood or PBMCs have shown that the number, nature, and location of negative charges strongly modulate the molecular conformation of endotoxin and biologic activity. Whereas monophosphorylated Lipid A exerts approximately 60-fold lower activity to induce IL-6 in blood cells, the phosphate-free Lipid A (e.g., dephosphorylated LPS of E. coli or phosphate-free Lipid A) almost completely loses their activity [185]. Furthermore, the number of the acyl chains in Lipid A was shown to greatly impact the stimulation activity of *E.coli*-derived Lipid A. Whereas hexaacyl Lipid A exhibited full agonistic activity, pentaacyl or tetraacyl Lipid A from *E.coli* lost their agonistic activity but kept full antagonistic activity [186]. Consequently, Lipid A derivatives with four lipid chains have antagonistic activity for the TLRD/MD-2 complex because all four lipid chains are completely submerged inside the hydrophobic MD-2 pocket, but cannot provide a hydrophobic dimerization surface that can be used for interaction with TLR4 [183].

In addition to TLR4, the structures of several other human TLRs in conjunction with their physiological or synthetic ligands have been described. TLR2 is unique among human TLRs because it can form heterodimers with other TLRs, such as TLR1 and TLR6. The principal ligands of the TLR1–TLR2 complex are triacyl lipopeptides, whereas the interaction with diacyl lipopeptides is substantially weaker. By contrast, the TLR2–TLR6 complex is able to bind to diacyl lipopeptides. These lipoproteins and lipopeptides are functionally and structurally diverse bacterial proteins anchored to the membrane by two or three covalently attached lipid chains [183]. Notably, typical TLR2 activators, such as Lipoteichoic acid also have multiple negatively charged phosphate groups.

## 6. Discussion of an Integrated Mechanistic Model

Many of the substitutions and deletions in the NTD and in the RBD of the spike protein can be well correlated with the observed increased infectivity and transmissibility of the Omicron variant, and with the high evasion potential from therapeutic monoclonal antibodies and vaccine-induced polyclonal neutralizing antibodies. Although three mutations near the S1/S2 furin cleavage site were expected to favor cleavage, a substantially lower cleavage efficacy has been found for the Omicron variant compared to the Delta variant and also compared to the Wuhan wild-type virus, correlating with significantly lower TMPRSS2-dependent replication in the lungs, but not in upper airway tissue, and lower syncytium formation [3,9], and a switch in primary cellular uptake pathway from membrane-based to endosomal cathepsin L dependent uptake [187], and lower pathogenicity in animal studies [7] and clinics [8].

A lower virus replication in lungs together with a faster replication in the upper respiratory system, such as nasopharyngeal and bronchi, can to a large extent explain Omi-cron’s greater ability for transmission between people while apparently causing less frequent acute respiratory distress syndrome (ARDS) of the lungs and systemic symptoms of COVID-19. However, the molecular mechanisms responsible for the reciprocal changes in cellular tropism of Omicron regarding upper and lower respiratory system [3,9,187] are not sufficiently defined at this moment. The following hypothesis tries to connect the various findings into an integrated model to explain the molecular and cellular pathways for the changed cellular tropism, and lower pathogenicity of the Omicron variant in comparison to all previous VoCs.

The following paragraph summarizes the most relevant findings described in the previous sections:(1)Different components of SARS-CoV-2, in particular of the spike protein, have been demonstrated to activate TLRs, in particular TLR4 and TLR2.(2)Dimerization represents the general principle underlying the activation of TLRs, with activating PAMPs serving as molecular linkers promoting dimerization.(3)For dimerization, there is a minimal number of hydrophobic chains necessary which have to fit into hydrophobic pockets in order to provide sufficient hydrophobic interactions, as demonstrated for TLR4-MD-2, TLR2-TLR1, and TLR2-TLR6 complexes, respectively.(4)There is a common feature of the TLR activating viral glycoproteins (also including the SARS-CoV-2 spike protein) with all of them being membrane-bound proteins which contain hydrophobic domains necessary for fusion with the host cell membrane and having the potential to interact with the hydrophobic pockets of TLR complexes.(5)Negatively charged groups have been shown to be essential for dimerization, as illustrated for TLR4-MD-2/LPS complexes.(6)Interaction and dimerization of respective TLR complexes triggers the inherent downstream signaling pathways, mainly the NF-κB pathway.

Based on these findings, we have developed the following hypothesis:–Some hydrophobic domains of the SARS-CoV-2 spike protein can interact with the hydrophobic pockets of TLR-complexes leading to dimerization and activation. In particular, the hydrophobic six-helix bundle fusion core structure (6HB) in the post- fusion state of the SARS-CoV-2 spike protein can be hypothesized to fit into the hydrophobic pockets of MD-2-TLR4. Other hydrophobic domains of the spike protein, such as the three hydrophobic stretches in the S2 subunit of the trimer in prefusion state, may be speculated to fit for binding to TLR2-TLR1/6 complexes.–Distribution of charged amino acids can greatly affect binding to and dimerization of TLR complexes. The changed charge distribution on the Omicron spike protein with high accumulations of positively charged amino acid residues in the RBD and in the S2 subunit, together with loss of several negatively charged amino acids by substitutions in the Omicron spike protein, may prevent high affinity binding to the TLR complexes and/or insufficient dimerization of TLR complexes, leading to lower downstream signaling and lower pro-inflammatory activation, lower NF-kB activation, and related lower furin expression. Indeed, a lower NF-κB activation by the Omicron variant vs. a whole panel of previous variants including the D614G, Delta, Lambda, and Mu variant has been shown recently [187].

### How Will this Impact the Virus Replication, Cellular Tropism, and Pathogenicity?

The SARS-CoV-2 binds via RBD of the spike protein to its high affinity receptor on the host cells. Binding to ACE2 triggers a series of conformational changes in the S2 subunit, i.e., the proteolytic cleavage between S1 and S2 by TMRRSS2, exposing hydrophobic parts of S2 to fuse the viral membrane with the membrane of the host cell, and enabling the penetration of viral RNA into the host cell. Alternatively, the virus can be taken up via clathrin-coated pits into endosomes, where proteolytic cleavage is taken over by cathepsin L. Within the host cell, the viral RNA is translated into non-structural proteins (NSPs) with massive translation viral RNA into viral non-structural and structural proteins. This process occurs mainly in bubble-like structures (i.e., double-membrane vesicles, DMVs) after remodeling of the cell’s endoplasmatic reticulum (ER). The newly produced viral components assemble into complete virus particles which leave the cell via transition through the Golgi apparatus (Figure 6). Most likely at this prefinal step, the host cell protease furin mainly presents in the TGN and cleaves the spike protein at the polybasic PRRAR site, preparing the spike protein for uptake into the next host cell. The massive remodeling of the host cell ER and viral RNA replication and translation, together with the fusogenic activity of the SARS-CoV-2 spike protein leading to excessive syncytia formation, will lead to a damage of the infected cells and to DAMPs formation, which are expected to trigger the cellular PPP alert mechanism.

Furthermore, different parts of the SARS-CoV-2 virus themselves act as PAMPs, triggering activation of a variety of pattern recognition receptors, in particular TLRs, leading to excessive activation of their intrinsic signal mechanisms, in particular the NF-κB pathway. In this context, the differentiated impact of various TLRs has to be taken into consideration. Whereas single-stranded viral RNA of SARS-CoV-2 or their double-stranded replication intermediates are expected to activate TLR7/TLR8 and TLR3, respectively, there are so far no indications that this should differ qualitatively from similar processes after infection by other coronaviruses, including the four seasonal low pathogenic viruses. Indeed, analysis of available data and their correlation with COVID-19 severity have shown no correlation for TLR3, whereas TLR7 correlated only with moderate COVID-19 severity, in contrast to other TLRs where activation increased with COVID-19 severity [147]. This may be due to the lack of MyD88 activation by TLR3, and a rather balanced induction of TLR7 downstream signaling leading to both MyD88 proinflammatory and TRIF- induced IRF activation, but may also correlate with the generally relatively low expression levels of TLR3 and TLR7/8 (less than 20 or 35 nTPM, respectively) compared to TLR2 or TLR4 (more than 160 and 120 nTPM, respectively, see Figure 4).

In contrast to low pathogenic coronaviruses, SARS-CoV-2 induces often an exuberated hyper inflammatory signature with cytokine/chemokine storm, massive coagulation disturbances, and systemic pro-inflammatory status in the endothelium, correlating with massive M1/Th1 cytokine release, with pathological and clinical feature shared with other highly pathogenic acute RNA virus infections, such as SARS-CoV, MERS-CoV, H5N1, and (Spanish type) H1N1 [45]. The underlying mechanism of these exaggerated pro-inflammatory reactions may rely on an additional, unbalanced and excessive NF-κB pathway activation due to powerful additional upstream signal triggers, which seem to be common for a variety of highly pathogenic acute RNA virus infections. In this context, binding and activation of TLR2 and TLR4 may play a major role. Various components of the SARS-CoV-2 virus have been demonstrated to activate TLR2 and TLR4, with the majority of reports showing massive activation by the spike proteins (and few reports for the envelope protein), always associated by excessive NF-κB activation. The excessive NF-κB activation can result from three major routes, i.e., (1) from “physiological” TLR7 and TLR3 activation by single-stranded RNA and/or double-stranded intermediates in the infected cells, (2) from excessive ER remodeling in the infected virus producer cells (i.e., primarily ACE2/TPRSS2 positive cells), and (3) from additional TLR4 and/or TLR2 activation in infected or even in non-productively infected innate cells (e.g., macrophages, endothelial cells). In this context, the relative cellular and tissue distribution/expression of the different TLRs will be a deciding factor with regard to their impact on COVID-19 severity. According to the Human Protein Atlas, TLR4 and TLR2 are expressed at high levels on different types of macrophages (TLR2 and TLR4) and endothelial cells (TLR4), and TLR2 in alveolar type 2 cells, whereas only very low levels are expressed on alveolar type 1 cells. This correlates with reports showing that cathepsin L expression, but not TMPRRS2 expression, is particularly prominent in macrophages and endothelial cells, with a demonstrated correlation between circulating levels of cathepsin L and disease course and severity in patients with COVID-19 [188].

The activation of the various types of TLRs, dominated by the highly expressed TLR4 and TLR2 on macrophages and endothelial cells, will trigger via their intrinsic signaling pathways excessive gene expression for a broad range of pro-inflammatory cytokines and chemokines, adhesion molecules, and acute phase proteins. Furthermore, the highly activated NF-κB pathway, directly or via HIF-1α activation or via cytokine release (such as IL-12), is expected to stimulate furin expression in the cells. Whereas furin can be expressed in most tissues, the basic expression levels seem to be very low. Viral infections, cancer, hypoxia, HIF-1α, and cytokines (e.g., IL-12) have been found to significantly increase furin expression. Activation of the NF-κB pathway triggering HIF-1α expression and cytokine release may play a central role in stimulating furin expression. Since cleavage at the furin-like site is essential for highly efficient TMPRSS2-dependent membrane uptake of SARS-CoV-2, the positive feedback from TLR-NF-κB-HIF-1α/IL-12-furin activation may play a significant role in providing enough of the enzyme necessary for site specific S1/S2 cleavage.

In this context the amino acid changes in the Omicron variant must be analyzed. For various mutations in the non-structural proteins of Omicron, their involvement in reducing an excessive innate response is not expected. In contrast, the rather suppressing effects of non-structural SARS-CoV-2 proteins, including nsp1, nsp3, nsp6, nsp8, ORF3b, and ORF8 resulting in suppression of IFN response have been described [16,147,189,190,191,192,193].

Regarding other viral structural proteins, there is only one amino acid substitution in the envelope protein T9I. In contrast, there are more than 30 mutations in the spike protein in the Omicron variant. Whereas the increased infectivity, transmissibility, and escape from immune reactions can largely be explained by substitutions or deletions in the RBD and NTD, the lower S1/S2 cleavage at the furin-like site, correlating with lower (or almost absent syncitia formation), leading to a significantly changed cellular tropism and attenuated pathogenicity, cannot be explained from the amino acid substitutions in the S1/S2 region. In this context, the interaction and activation of the TLR2-TLR1/6 and TLR4-MD-2 receptors may be the missing link. Hydrophobic and polar-charged interactions of the archetypic TLR4 agonist, LPS, with the TLR4-MD2 complex may serve as a prototype for interaction with other ligands, including the SARS-CoV-2 spike protein. Interaction of the hydrophobic domains of the SARS-CoV-2 spike protein with hydrophobic pockets, e.g., in the MD2-TLR4 or TLR2-TLR1/6, will depend on hydrophobic structures and appropriately localized and charged amino acids necessary for efficient interaction with TLR-complexes and their dimerization. Typical agonists for TLR4-MD-2 and TLR2, such as Lipid A or Lipoteichoic acid, respectively, have typical hydrophobic chains linked to exposed negatively charged phosphate groups.

We speculate that the hydrophobic six-helix bundle fusion core structure (6HB) in the post-fusion state of the SARS-CoV-2 spike protein with the central coiled-coil fusion core formed by the three HR1 domains surrounded by the three HR2 provides hydrophobic bundle-like structures, and resembling to some extent the hydrophobic structure of fatty acids in Lipid A, may be able to interact with the TLR4-MD-2 complex. Alternatively, hydrophobic stretches on the trimer in the pre-fusion state may fit into the hydrophobic pockets of MD-2-TLR4 or TLR2-TLR1/6 complexes, respectively, leading to dimerization and triggering the TLR-typical downstream signaling pathways. In this context, the significant change in the charge distribution in the Omicron spike protein, with multiple additional positively charged amino acid substitutions accompanied by loss of several negatively charged amino acids, and in the molecular vicinity of hydrophobic 3 or 6 bundled coils, may prevent high affinity binding to the TLR complexes and/or insufficient dimerization of TLR complexes.

Although probably only some of the TLR, i.e., TLR4 and/or TLR2, pathways are primarily affected by the charged electrostatic charge pattern of Omicron, this shortfall likely concerns the most powerful pathways, because of the high relative expression of TLR2 and TLR4 on innate inflammatory cells and the exceptionally high pro-inflammatory capacity of the affected innate immune cells, in comparison to the primarily SARS-CoV-2 infected bronchial or alveolar cells. This reduced innate immune cell activation, leading to lower NF-κB and HIF-1α activation, counterbalances the advantage from the preserved (or even slightly enhanced) polybasic furin cleavage site in the spike protein of Omicron because of the limited proprotein convertase (furin) availability, which may explain to a large extent the changed cellular tropism and the lower pathogenicity of the Omicron variant.

## 7. Conclusions

The recently appearing Omicron variant shows surprising reciprocal changes in cellular tropism of the regarding upper and lower respiratory system, which cannot be explained simply by the changed binding to the ACE2 receptor or immune escape. Here we present a new hypothesis proposing that the changed distribution of charged amino acids in the spike protein of the Omicron variant compared to all other VoCs may disturb the recognition by innate Pattern-recognition receptors (PRRs), in particular of certain Toll-like receptors (TLR), resulting in lower activation of the NF-κB pathway and related signaling pathways, and also resulting in lower furin expression, lower viral replication in the lungs, and lower systemic immune hyperactivation.

## Figures and Tables

**Figure 1 ijms-23-05966-f001:**
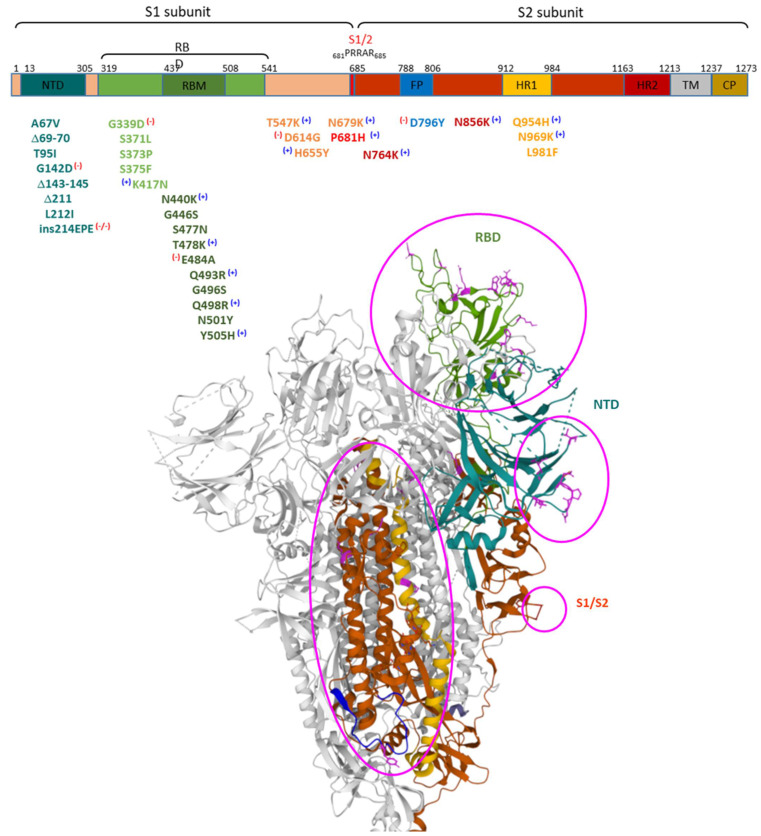
Structure of the SARS-CoV-2 spike protein and mutations in the Omicron variant and spatial illustration of mutated sites based on the published cryo-structure PDB 7TEI SARS-CoV-2 Omicron 1-RBD up Spike Protein Trimer. PDB DOI: 10.2210/pdb7TEI/pdbEM Map EMD-25846: EMDB EMDataResource [66]. Protomer 1—brown, except RBD—light green, NTD—dark turquoise (ride side), HP1—yellow, FP—blue, Omicron mutations—pink, Protomers 2 & 3—grey. Regions of the Omicron spike protein with higher number of mutations are circled in pink.

**Figure 2 ijms-23-05966-f002:**
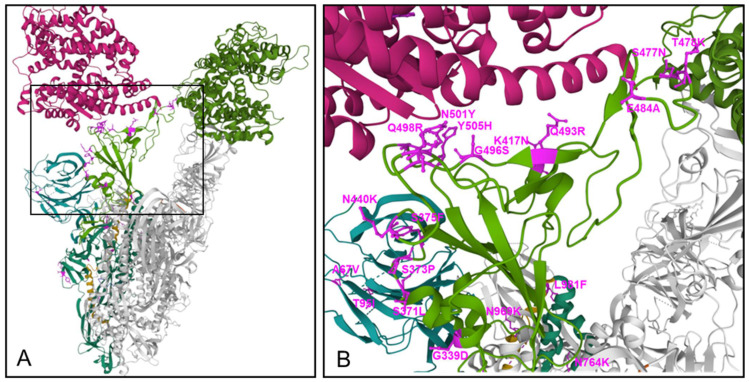
The Omicron spike protein trimer, binding to two ACE2 molecules (**A**), and the RBD—ACE2 binding interface (**B**) are shown, derived from the PDB **7T9K** [67]. Protomer 1—dark green, except RBD—light green, binding to one ACE2 molecule (dark red, **upper left corner**), NTD—dark turquoise (**left side**), HP1—yellow, Omicron mutations—pink, Protomers 2 and 3—grey, with one of them binding to another ACE2 (middle green, **upper right corner** (**A**)).

**Figure 3 ijms-23-05966-f003:**
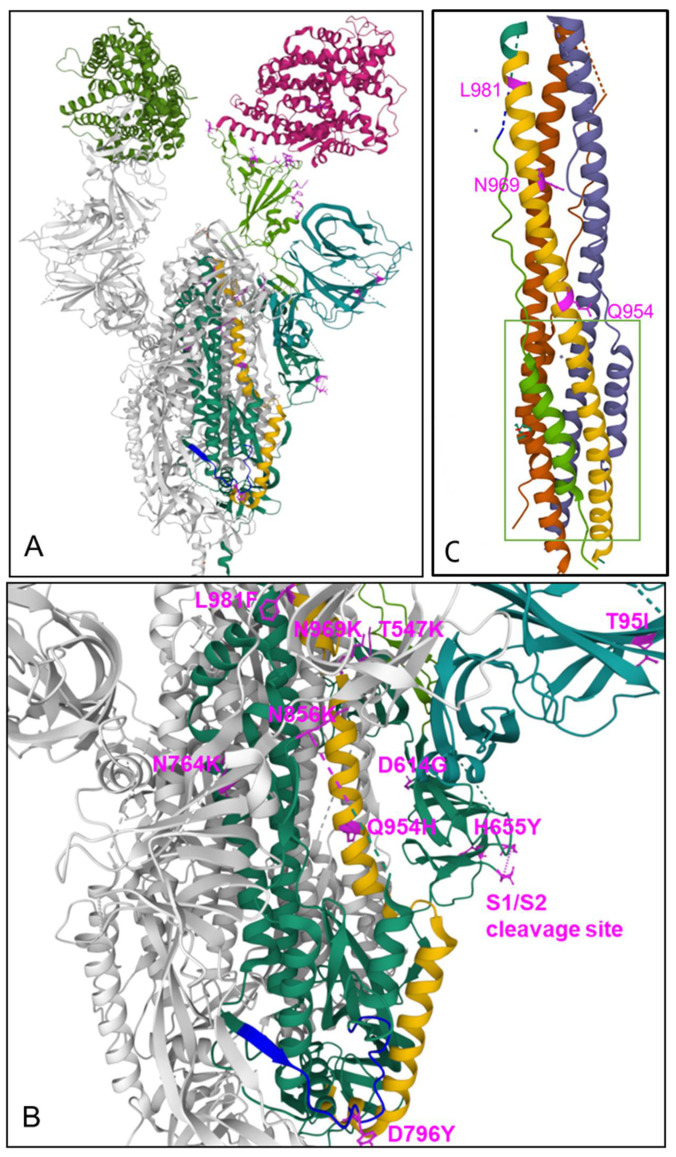
The Omicron spike protein trimer, binding to two ACE2 molecules (**A**), and S2 subunit (zoomed) (**B**) are shown, derived from the PDB **7T9K** [67]. Protomer 1—dark green, exept RBD—light green, binding to one ACE2 molecule (dark red, **upper right corner, A**), NTD—dark turquoise (**ride side**), HP1—yellow, FP—blue, Omicro mutations—pink, Protomers 2 & 3—grey, with one of them binding to another ACE2 (middle green, **upper left corner, A**), Two substitutions within the S1/S2 cleavage site (i.e., N679K, P681H) are not displayed in this model, (**C**) PDB 6LXT Structure of post-fusion core of wild-type 2019-nCoV S2 subunit [19]. The 6HB bundle highlighted by the green frame.

**Figure 4 ijms-23-05966-f004:**
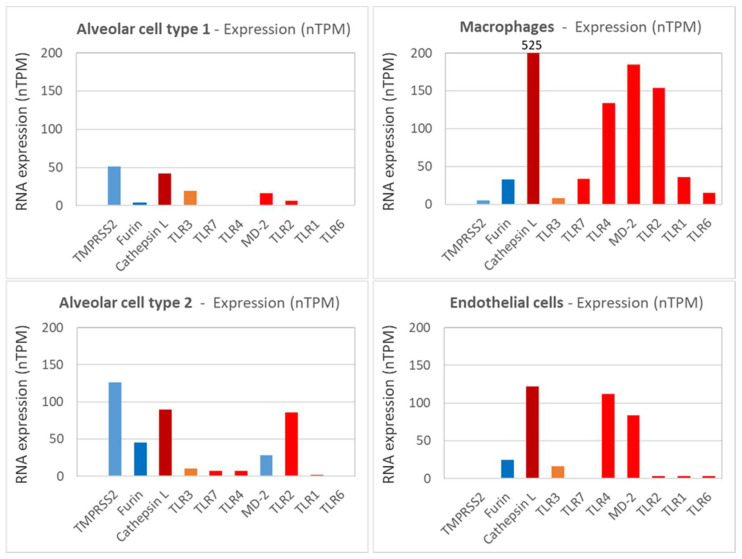
The expression of TMPSS2, cathepsin L, furin, TLR1, 2, 3, 4, 6, 7, and MD-2 normalized to nTPM (i.e., Transcripts per million protein coding genes) for representative cell types from the lungs are shown. Data are derived from the Human Protein Atlas [119].

**Figure 5 ijms-23-05966-f005:**
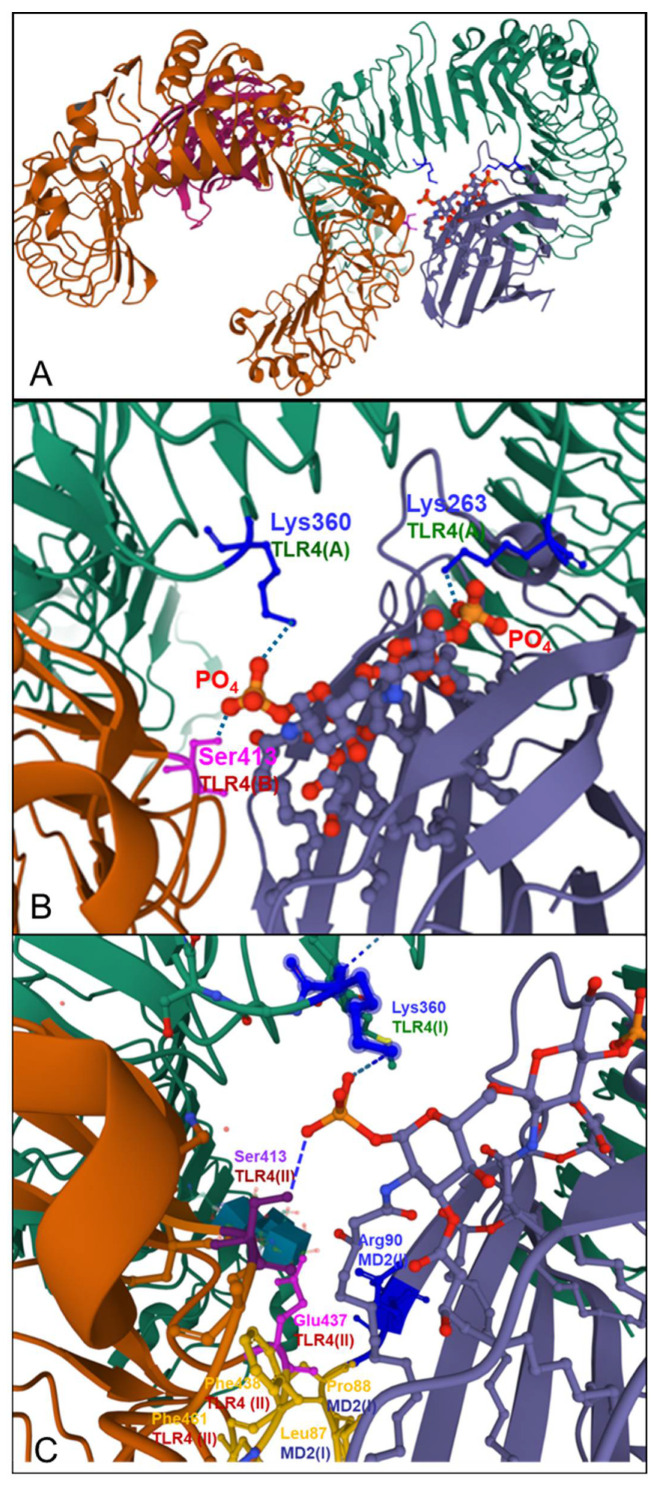
Dimerization of TLR4/MD-2 complexes by LPS is shown based on RCSB PDB 3VQ2 Crystal structure of mouse TLR4/MD-2/LPS complex (DOI: 10.2210/pdb3VQ2/pdb [184]. (**A**) At higher magnification: Insertion of LPS (blue/red ball stick model) into the MD-2 hydrophobic pocket (**blue**) and charged and polar interaction of the negatively charged phosphate groups of Lipid A with Lys360 and Lys263 of TLR4(I) (**green**), (**B**) and with Ser413 of the second TLR4(II) (**brown**) (**B**,**C**) and hydrophobic amino acid residues (**yellow**) in MD-2(I) (blue) and TLR4(II) (**brown**), (**C**) in the dimerization interface are depicted.

**Figure 6 ijms-23-05966-f006:**
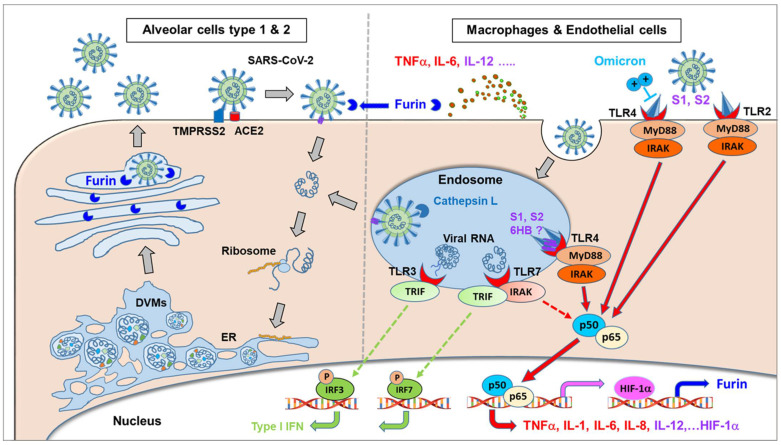
The SARS-CoV-2 binds via RBD of the spike protein to ACE2. Binding triggers proteolytic cleavage between S1and S2 by TMPRRS2, resulting in formation of 6HB and fusion with the host cell membrane allowing penetration of viral RNA into the host cell. Viral RNA transcription and translation of viral non-structural and structural proteins occurs in double-membrane vesicles (DMVs) after remodeling of the endoplasmatic reticulum (ER). The newly produced viral components assemble into complete virus particles which leave the cells via the Golgi apparatus where the spike protein undergoes proteolytic cleavage by furin. This process preferably occurs in virus producer cells with high TMPRSS2 expression, e.g., alveolar cells (left side). Alternatively, the virus can be taken up via clathrin-coated pits into endosomes, where proteolytic cleavage is taken over by cathepsin L. The endosomal uptake is predominant in TMPRSS2-negative cells, but not in cathepsin L-rich cells, such as innate immune cells and endothelial cells (right side). Several components of the SARS-CoV-2 virus act as PAMPs activating various Toll-like receptors (TLRs), resulting in massive activation of the NF-kB (p50/p65) pathway. The relative cellular distribution/expression differs greatly for the different TLRs with high levels of TLR4 and TLR2 on macrophages and endothelial cells, whereas only low levels are expressed on alveolar lung cells. The activation of TLRs will trigger activation of NF-κB pathway, which triggers HIF-1α activation and expression of cytokine, including TNFα, IL-1, IL-6, and IL-12. HIF-1α and IL-12 have been shown to increase furin expression. As a hypothesis for TLR activation, interaction of hydrophobic domains (and a distinct charged amino acid pattern) of the SARS-CoV-2 spike protein may be necessary for dimerization and activation of MD2-TLR4 or TLR2-TLR1/6, triggering the TLR-typical downstream signaling pathways, as has been shown so far for the whole spike protein, S1 and S2. As a hypothesis (?), the hydrophobic six-helix bundle fusion core structure (6HB) in the post-fusion state of the SARS-CoV-2 spike protein may fit into the hydrophobic pockets of MD-2-TLR4 or TLR2-TLR1/6 complexes, respectively. The significant change in the charge distribution in the Omicron spike protein, with multiple additional positively charged amino acid substitutions may prevent high affinity binding to the TLR complexes and/or insufficient dimerization of TLR complexes, leading to lower NF-kB signaling with lower expression of cytokines, HIF-1α, resulting in lower furin expression and insufficient S1/S2 cleavage, despite the presence of the polybasic furin cleavage motif.

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
