# Peer review of "Could a Lower Toll-like Receptor (TLR) and NF-κB Activation Due to a Changed Charge Distribution in the Spike Protein Be the Reason for the Lower Pathogenicity of Omicron?"

_ijms, 2022, doi:10.3390/ijms23115966_

Round 1

Reviewer 1 Report

The review is very interesting and definitely deserves publication. The manuscript could be accepted in its present form.

Author Response

We want to thank the reviewer for the encouraging assessment. Some minor changes have been introduced in the revised manuscript based on the recommendations of the reviewers. A revised manuscript will be uploaded.

Reviewer 2 Report

The authors Ralf Kircheis and Oliver Planz in the review “Could a lower Toll-like receptor (TLR) and NF-kB activation due to a changed charge distribution in the spike protein be the reason for the lower pathogenicity of Omicron?” hypothese that the changed charge distribution in the Omicron’s spike protein could lead to lower activation of TLRs in innate immune cells resulting in lower NF-kB activation, furin expression, and viral replication in the lung, and lower immune hyper-activation. The review, whose topic is very interesting, is written clearly and in detail. The first part of the work, supported by clear figures, describes the mutations of the spike protein in the Omicron variant and is easy to read and understand. The second part of the review, which describes how SARS-CoV2 affects the NF-kB pathway is more complex. Paragraphs 4.2 and 4.4 should be simplify and the paragraph 5.5 should be deleted or synthesized/incorporated into the text. Finally, the discussion and conclusions are well written and the model presented is very interesting.

Line 570: replace actiaction with activation.

Author Response

We want to thank the reviewer for the encouraging assessment and the recommendations. We have addressed all recommendations in the revised version of the manuscript uploaded.

In detail: paragraphs 4.2 and 4.4 have been simplified and shortened . Paragraph 5.5 was incorporated into paragraph 5.4.

Reviewer 3 Report

In their work, the authors take an in-depth look at SARS-CoV2 and its disease.

They describe in detail the molecular and cellular mechanisms that are triggered by infection and try to show why the omicron variant differs from its predecessors. Based on the molecular data, which are summarized here in detail, they hypothesize what effects the mutations in the spike protein have on the activation of TLRs and NF-KB.  The work is very comprehensive and presents a good summary of the data to date in a condensed well-linked manner. 

Some minor points the authors should change/reconsider:

1.)Figure caption is not uniform Fig.1, Figure 2

Figure 1 : Labelling is missing for the central helices that are circled
Figure 2 is never mentioned in the text
Figure 4: should be redesigned. Deviation should be added and probably a box/whiskers plot would fit better here. y-axis is not labelled
Figure 6 should be closer to the text where it is mentioned

2.) Commas and hyphens missing throughout the text

Line 633: change HIF-1A to α
Line 838: citations are written in bold here
Line 1068 : variant(t)
Line 1133: has be considered
check the text for typos
et.al. should be in italics 
6.Discussion --> altered font size?

3.) The authors neglected the fact that there is another co-receptor for SARS-CoV2 spike protein present on the cell surface. Due to its high negative charge it can also contribute to better binding of the "more positive" Omicron RBD. 

4.) Line 408 : Clinical trial accession numbers should be added to understand what exactly is referred here

5.) Lines 1071-1107 should be restructured. For the reader, it is not easy to follow the exact train of thought of the authors. Which hypotheses are based on which statements? This paragraph should be restructured as it is a central part of the work 

6.) A short conclusion should be added that summarizes the main points of the review

Author Response

We want to thank the reviewer for the encouraging assessment and the recommendations. We have addressed all recommendations in the revised manuscript uploaded.

In detail:

Reg 1) Alle minor points have been corrected: including Figure captures, labelling Fig. 1, reference of Fig. 2 in the text.

Figure 4 has been improved by adding y-axis label. Unfortunately, since the data are derived from scatter plots provided in Human protein atlas web site, no deviations (e.g. in a box/wiskers plot) can be calculated and shown

Fig. 6 has been shifted more closely to the site of first reference in the text

Reg 2) Commas and hyphens have been corrected throughout the text (and may be also additionally subject to final editing by the publisher.

All meansioned minor corrections have been introduced.

Reg. 3) the potential effect of the changed charge distribution (higher positive charge) in the Omicron spike protein on binding to various (including putative) co-receptors with highly negative charge (such as heparane sulfate and/or sialic acid) has been included in 3.1.. A new reference Nr. 11 has been included.

Reg 4) References have been included

Reg 5) the paragraph has been reconstructed and the text optimized to make the main message more clear

Reg 6) a short conclusion has been added